



# Data Assimilation of Volcanic Aerosols using FALL3D+PDAF

Leonardo Mingari[1], Arnau Folch[2], Andrew T. Prata[3], Federica Pardini[4], Giovanni Macedonio[5], and
Antonio Costa[6]

[1]Barcelona Supercomputing Center, Barcelona, Spain
[2]Geociencias Barcelona (GEO3BCN-CSIC), Barcelona, Spain
[3]Sub-department of Atmospheric, Oceanic and Planetary Physics, University of Oxford, Oxford, U.K.
[4]Istituto Nazionale di Geofisica e Vulcanologia, Sezione di Pisa, Pisa, Italy
[5]Istituto Nazionale di Geofisica e Vulcanologia, Osservatorio Vesuviano, Naples, Italy
[6]Istituto Nazionale di Geofisica e Vulcanologia, Sezione di Bologna, Bologna, Italy

**Correspondence:** leonardo.mingari@bsc.es (LM)

**Abstract.** Modelling atmospheric dispersal of volcanic ash and aerosols is becoming increasingly valuable for assessing the potential impacts of explosive volcanic eruptions on infrastructures, air quality, and aviation. Management of volcanic risk and reduction of aviation impacts can strongly benefit from quantitative forecasting of volcanic ash. However, an accurate prediction of volcanic aerosol concentrations using numerical modelling relies on proper estimations of multiple model parameters
which are prone to errors. Uncertainties in key parameters such as eruption column height, physical properties of particles or meteorological fields, represent a major source of error affecting the forecast quality. The availability of near-real-time geostationary satellite observations with high spatial and temporal resolutions provides the opportunity to improve forecasts in an operational context by incorporating observations into numerical models. Specifically, ensemble-based filters aim at converting a prior ensemble of system states into an analysis ensemble by assimilating a set of noisy observations. Previous studies dealing
with volcanic ash transport have demonstrated that a significant improvement of forecast skill can be achieved by this approach. In this work, we present a new implementation of an ensemble-based Data Assimilation (DA) method coupling the FALL3D dispersal model and the Parallel Data Assimilation Framework (PDAF). The FALL3D+PDAF system runs in parallel, supports online-coupled DA and can be efficiently integrated into operational workflows by exploiting high-performance computing (HPC) resources. Two numerical experiments are considered: (i) a twin experiment using an incomplete dataset of synthetic
observations of volcanic ash and, (ii) an experiment based on the 2019 Raikoke eruption using real observations of $SO_2$ mass loading. An ensemble-based Kalman filtering technique based on the Local Ensemble Transform Kalman Filter (LETKF) is used to assimilate satellite-retrieved data of column mass loading. We show that this procedure may lead to nonphysical solutions and, consequently, conclude that LETKF is not the best approach for the assimilation of volcanic aerosols. However, we find that a truncated state constructed from the LETKF solution approaches the real solution after a few assimilation cycles,
yielding a dramatic improvement of forecast quality when compared to simulations without assimilation.



# 1 Introduction

Volcanoes encompass a range of hazardous phenomena that precede, accompany, and follow volcanic eruptions. Fragmented magma and gases released during explosive eruptions raise up to a neutral buoyancy level where volcanic aerosols and ash can be transported thousands of kilometres by upper-level winds. Specifically, volcanic ash clouds jeopardise flight safety, whereas the subsequent ash fallout can affect buildings, infrastructures, communication networks, airports, power plants, and water and energy distribution networks (Wilson et al., 2014; Clarkson et al., 2016). Management of volcanic risk and related strategies for reducing its impacts on aerial navigation can benefit from accurate forecasts of volcanic dispersal produced by Volcanic Ash Transport and Dispersion (VATD) models (e.g., Folch, 2012). For example, operational institutions like the Volcanic Ash Advisory Centers (VAACs) rely on VATD models to deliver volcanic ash forecasts to aviation stakeholders, civil protection agencies, and governmental bodies (e.g., Beckett et al., 2020). VATD models aim at simulating the main processes involved in the life cycle of atmospheric ash and gas species released during volcanic eruptions: emission, atmospheric transport, and ground deposition.

The accuracy of forecasts depends on multiple factors involving model spatial resolution, under-laying meteorological driver, model physics and related parameterisations, or uncertainties on Eruption Source Parameters (ESP), e.g., column height, mass eruption rate, particle size distribution, and vertical mass distribution. In fact, uncertainties in ESP are known to be first-order contributors to model errors (Costa et al., 2016b; Poulidis and Iguchi, 2021). Additionally, in order to properly define the emission source term for complex plume dynamics, models require time-varying ESP (e.g., Suzuki et al., 2016b), which are typically poorly constrained during eruptive scenarios. Data assimilation (DA) is one of the most effective ways to reduce forecast errors through the incorporation of observation data into numerical models (e.g., Kalnay, 2003). In an assimilation step, a forecast is used as first guess to obtain an improved estimate of the system state by incorporating the available observations. The estimate of an initial state to start a forecast system applying DA techniques is a well-established practice in numerical weather prediction, widely used in research (e.g., Anderson et al., 2009) and operations (e.g., Whitaker et al., 2008; Kleist et al., 2009; Bonavita et al., 2016). Specifically, the Ensemble Kalman filter (EnKF) has been widely used in oceanographic and atmospheric sciences for performing 4D data assimilation (Evensen, 2003). Ensemble data assimilation attempts to represent the error statistics using an ensemble of model states instead of storing the full covariance matrix (e.g., Carrassi et al., 2018).

Similarly, it is long recognised that forecasting of volcanic clouds using VATD models can benefit from remote sensing observations (Bonadonna et al., 2012). Previous work has already demonstrated that a substantial improvement of quantitative ash forecasts can be achieved by using ensemble-based data assimilation methods (Fu et al., 2015, 2016, 2017b,a; Osores et al., 2020; Pardini et al., 2020). However, transfer of DA techniques into operational environments is yet limited, partly because these approaches require of multiple model runs to generate an ensemble of forecasts, making high-resolution modelling challenging under time-constrained operational contexts, particularly if computational resources are limited. Additionally, four-dimensional variational data assimilation (4D-Var) methods have been proposed for the reconstruction of the vertical profile of volcanic eruptions (Lu et al., 2016a,b).





The emergence of near-real-time geostationary satellite measurements with high spatial and temporal resolutions provides
the opportunity to improve the accuracy of operational forecasts. With last-generation satellite instrumentation, observations
can be available every 10–15 minutes at 2–4 km pixel size. For example, the Spinning Enhanced Visible and Infra-Red Imager
(SEVIRI) on board of the Meteosat Second Generation (MSG) platform provides observations of the full disk with 3 km
resolution at the sub-satellite point for all channels (except for the high-resolution visible channel) in observation intervals
of 15 minutes for full disk (Schmetz et al., 2002). Similarly, the Advanced Himawari Imager (AHI) instrument aboard the
Himawari-8 geostationary satellite (Bessho et al., 2016) samples the Earth's full disk every 10 minutes with a spatial resolution
of 2 km at the sub-satellite point for the infrared channels.

Recently, the FALL3D code (Folch et al., 2009) has been redesigned and rewritten in the framework of the EU Center of
Excellence for Exascale in Solid Earth, *ChEESE*. The code version 8.0 (Folch et al., 2020; Prata et al., 2021) is tailored to
extreme-scale computing requirements and presents substantial improvements on code scalability, computational efficiency,
memory management, and overall capability to handle much larger problems. In addition, the code version 8.1 (Folch et al.,
2021) implemented ensemble forecast capabilities and validation metrics. New developments have led to improved quality
of forecasts, enabled to quantify model uncertainties, and laid the foundations for the incorporation of ensemble-based DA
techniques into future releases of FALL3D.

This work presents a new data assimilation system based on the coupling between FALL3D and the Parallel Data Assimila-
tion Framework (PDAF, Nerger et al., 2005, 2020), available in the last code release (version 8.2) of FALL3D. The proposed
methodology can be efficiently implemented in operational environments by exploiting High Performance Computing (HPC)
resources. The FALL3D+PDAF system can run in parallel and supports online-coupled DA, which allows an efficient data
transfer management through parallel communications among the ensemble members. The main objective of this paper is to
present and validate an ensemble-based data assimilation system suitable for efficient implementation in operational workflows
by exploiting HPC capabilities. The proposed methodology aims at producing a substantial improvement in quantitative fore-
casting of volcanic aerosols taking advantage of high-resolution retrievals from the new generation of satellite instrumentation.
The evaluation of the DA system comprises two numerical experiments using the Local Ensemble Transform Kalman Filter
(LETKF, Ott et al., 2004; Hunt et al., 2007). Firstly, we propose a twin experiment using a dataset of synthetic observations
based on an idealised volcanic eruption. In this case, the observational dataset is defined using noisy mass loading (i.e., to-
tal column mass per unit area) data of volcanic ash. Secondly, we simulate the 2019 Raikoke volcanic eruption considering
satellite-retrieved mass loading of $SO_2$ for assimilation purposes.

The manuscript is organised as follows. A description of the FALL3D+PDAF DA system is outlined in Sect. 2. Section 3
presents the aforementioned numerical experiments to evaluate the performance of the modelling system under different con-
figurations. Results of the experiments are discussed in Sect. 4 and recommendations are made concerning future studies.
Conclusions are drawn in the final Sect. 5.



## 2 Data Assimilation System

An online DA system has been implemented in the latest version release of FALL3D (v8.2), an open-source code with an active community of users worldwide. FALL3D is an Eulerian model for atmospheric passive transport and deposition based on the so-called Advection-Diffusion-Sedimentation (ADS) equation (Folch et al., 2020). The code has been redesigned and rewritten from scratch in the framework of the EU Center of Excellence for Exascale in Solid Earth (*ChEESE*) in order to overcome legacy issues and allow for successive optimisations in the preparation towards extreme-scale computing. The new versions include significant improvements from the point of view of model physics, numerical algorithmic methods, and computational efficiency. In addition, the capabilities of the model have been extended by incorporating new features such as the possibility of running ensemble forecasts and dealing with multiple atmospheric species (i.e., volcanic ash and gases, mineral dust, and radionuclides). Efforts to implement ensemble capabilities on the previous release of FALL3D (v8.1) not only made it possible to quantify model uncertainties and improve forecast quality (Folch et al., 2021) but also paved the way for ensemble-based data assimilation techniques to be integrated into subsequent versions of FALL3D.

### 2.1 FALL3D+PDAF

Data assimilation (DA) techniques are extensively used to study and forecast geophysical systems and can be applied to a broad range of operational and research scenarios (Carrassi et al., 2018). Generally speaking, DA techniques aim at obtaining an optimal state of a dynamical system by combining model forecasts with observations using either a sequential or a variational method. In sequential schemes, the assimilation process is characterised by a sequence of steps involving a forecast step and a subsequent analysis in which the *a posteriori* estimate is obtained from the *a priori* forecast state by incorporating observational information. Ensemble-based methods are a family of algorithms providing a practical method to deal with realistic high-dimensional geophysical problems by means of a low-dimensional approximation of the background error covariance. The ensemble data assimilation is based on the generation of an ensemble of trajectories of the model dynamics. An ensemble of forecasts provides information about uncertainty in prior forecast at the analysis time. Observations are assimilated in the analysis step to generate a posterior ensemble defining an ensemble of initial conditions which are further propagated in time by the physical model until the next analysis time. This sequential data assimilation approach is represented in Fig. 1.

In this work, the model state for each ensemble member is propagated by the FALL3D dispersal model. The DA system builds upon an efficient implementation by coupling FALL3D and the Parallel Data Assimilation Framework (PDAF), an open-source software environment for ensemble data assimilation providing fully implemented and optimised data assimilation algorithms, including ensemble Kalman filters (KF) such as EnKF, ETKF, and LETKF (Nerger et al., 2005, 2020, see also Sect. A). PDAF supports an efficient use of parallel computers and facilitates its implementation by combining an existing numerical model with a group of DA algorithms with minimal changes in the model code. We used the PDAF version 1.14 that, in addition to KF algorithms, includes also an ensemble square root filter for nonlinear data assimilation, referred as the nonlinear ensemble transform filter (NETF, Tödter and Ahrens, 2015), and a Particle Filter (PF, e.g., Gordon et al., 1993).





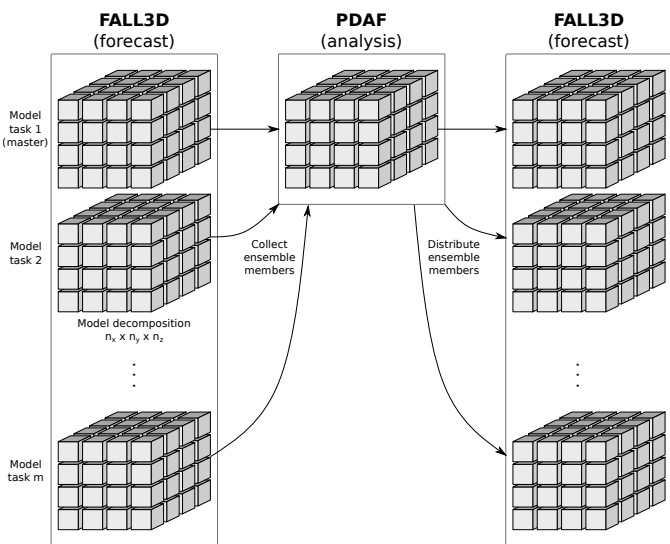

**Figure 1.** Scheme of the ensemble-based data assimilation system implemented in the FALL3D dispersal model. The system builds upon an efficient implementation by coupling FALL3D and the Parallel Data Assimilation Framework (PDAF) and uses a two-level parallelisation scheme based on MPI (Message Passing Interface).

The FALL3D+PDAF system can be run in parallel and supports online-coupled DA, enabling the workflow to be executed in a single step and with an efficient data transfer management through parallel communications. This avoids the creation of extremely large files that would be required to store the full system state in case of an off-line approach. The implementation uses a two-level parallelisation scheme based on MPI (Message Passing Interface) and can benefit from high-performance computing (HPC) resources. The two-level parallelisation scheme is sketched in Fig. 1. During the ensemble forecast phase, $m$ FALL3D model tasks run concurrently as an embarrassingly (or perfectly) parallel workflow to evolve the member states in time (level 1). In turn, each model task is executed by a single parallel instance of FALL3D, which uses a three-dimensional domain decomposition with $n_x$, $n_y$, and $n_z$ sub-domains along each direction (level 2). Consequently, the ensemble forecast requires a total of $m \times n_x \times n_y \times n_z$ MPI ranks. Multiple intra-member (level 1) communications are required during each assimilation step in order to collect and distribute the state vectors between different parallel tasks. Specifically, model tasks communicate with the master model task (i.e., the 1st model task in Fig. 1) during the analysis stage and filter operations are performed exclusively by ranks corresponding to the master model task.

## 2.2 Data assimilation algorithm

Two ensemble Kalman filter algorithms have been implemented in the FALL3D+PDAF system: ETKF and LETKF. Appendix outlines the theoretical background for these methods and the Kalman filter theory (see Sect. A). The Ensemble Transform Kalman Filter (ETKF, Bishop et al., 2001) is a popular square-root filter formulation which provides a practical method for data assimilation suitable for high-dimensional systems, relatively easy to implement and computationally efficient. However,





this work focuses exclusively on the localised version of the ETKF (i.e., LETKF) proposed by Hunt et al. (2007), which can provide more realistic results for volcanic aerosols than its global counterpart, ETKF.

Local analysis is performed by computing a separate analysis for each local domain and considering only observations within a volume defined by a cylinder of radius ($L_R$). No vertical localisation is used since observations are column integrated. Localisation in LETKF is performed by partitioning the state vector into a number of local domains defined by the vertical

column corresponding to a single cell of the horizontal model grid, and includes the variables corresponding to all bin species contributing to the observed column mass. A separate analysis is then generated for each model grid point in the local domain. By default, a uniform weight (unit weight) is assumed for all observations contributing to the local analysis. Alternatively, the influence of observations can also decay exponentially with the distance $r$ from the analysis location according to a weight with the dependency $\exp(-r/L_{SR})$, where the exponential decay radius, $L_{SR}$, is a user-defined input.

Table 1 lists the parameters required by the FALL3D input file to configure the data assimilation system. In addition to start/end time and frequency of assimilation, local range and inflation factor can be defined in this block. Note that the co-variance inflation factor is expressed here in terms of the so-called forgetting factor, defined as $\rho = \lambda^{-1} \leq 1$ (Nerger et al., 2012, see also Sect. A). Other parameters include satellite filename, type of observation weighting and cut-off diameter for volcanic ash (i.e., maximum particle diameter to compute mass loading). The parameter TRANSFORMATION specifies

how the $\boldsymbol{\Lambda}$ matrix, defined in Appendix by Eq. (A10), is computed conforming to two possible transformation options: identity matrix (DETERMINISTIC) or random rotation (RANDOM_ROTATION). As explained below in Sect. 2.3, the parameter SQRT_TRANSFORMATION allows the user to specify whether a nonlinear transformation should be applied to the model state variable.

Alternative ensemble-based techniques provided by PDAF, such as PF and NETF (see Sect. 2.1), will be implemented in

future releases of FALL3D. While the ensemble Kalman filters implicitly assume that the prior state and the observation errors are Gaussian, NETF and PF methods are not restricted by the assumptions of linearity or Gaussian noise. In contrast, PF and NETF are exposed to weight collapse due to the so-called curse of dimensionality (e.g., Carrassi et al., 2018). In addition, Kalman filters are expected to outperform NETF and PF in a linear and Gaussian problem (e.g., see Tödter and Ahrens, 2015). FALL3D solves an almost linear problem with weak non-linearity effects (e.g., due to gravity current, wet deposition, or

aggregation). However, as discussed next, the Gaussian hypothesis is not fulfilled, leaving open the question of which is the best approach to deal with the assimilation of volcanic aerosols.

### 2.3 Model state

The DA algorithm requires a model state vector $\boldsymbol{x} \in \mathbb{R}^n$ which is corrected in the analysis step. The state vector is constructed from the three-dimensional concentrations $C_i(x,y,z,t)$ at the assimilation time $t$ for the bin species $i$ ($i = 1, 2, \dots$). As con-

centration is a positive-semidefinite variable, the prior PDF associated with the ensemble forecast tends to show a right-skewed distribution. To illustrate this aspect, the two-dimensional histogram in Fig. 2a shows the skewness $\tilde{\mu}_3$ (i.e., $\mu_3/\sigma^3$, the third standardised moment) of the prior PDF computed for each grid cell at the first assimilation time for the Raikoke experiment (see Sect. 3.2). Note that a positive skewness ($\tilde{\mu}_3 > 0$) predominates in all points, with the most probable value ($\tilde{\mu}_3 \approx 11$)





**Table 1.** List of input parameters required by the data assimilation block in the FALL3D input configuration file

| Parameter | Options | Description |
| --- | --- | --- |
| `ASSIMILATION` | `ON/OFF` | Enable assimilation |
| `FILTER` | `ETKF/LETKF` | Type of filter |
| `ASSIMILATION_START` | Float value | Assimilation start time |
| `ASSIMILATION_END` | Float value | Assimilation end time |
| `FREQUENCY` | Float value | Assimilation frequency in hours |
| `FORGETTING_FACTOR` | Float value | Forgetting factor $\rho \in (0,1]$ |
| `LOCAL_RANGE` | Float value | Local radius for observations $(L_R)^{\dagger}$ |
| `TRANSFORMATION` | `DETERMINISTIC/RANDOM_ROTATION` | Type of ensemble transformation |
| `WEIGHTING` | `UNIFORM/EXPONENTIAL` | Observation weighting |
| `SUPPORT_RANGE` | Float value | Exponential decay radius $(L_{SR})^{\dagger}$ |
| `SATELLITE_FILE` | Filename | Input file with observations in netCDF format |
| `SATELLITE_DICTIONARY_FILE` | Filename | Input table with netCDF variables |
| `ASSIMILATED_TRACER` | `TEPHRA/SO2/H2O` | Species to assimilate |
| `DIAMETER_CUT_OFF` | Float value | Cut-off diameter for volcanic ash in μm |
| `IGNORE_ZEROS` | `YES/NO` | Ignore non-positive observations |
| `SQRT_TRANSFORMATION` | `YES/NO` | Apply a square root transformation to $\boldsymbol{x}$ |

$^{\dagger}$ $L_R$ and $L_{SR}$ are defined in units of the model grid size

occurring when the mean-to-sigma ratio (i.e., $\mu/\sigma$, the mean to standard deviation ratio) approaches to zero. Interestingly, the
relationship $\tilde{\mu}_3 = \sigma/\mu$ (solid red line) defines a lower boundary which is satisfied for almost all points ($\tilde{\mu}_3 > \sigma/\mu$). The skew-
ness of the a prior PDF tends to the expected value for a normal distribution ($\tilde{\mu}_3 = 0$) only for large values of $\mu/\sigma$. However,
values of $\mu/\sigma$ above 0.5 are extremely unlikely to occur and, in general, skewness values satisfy $\tilde{\mu}_3 > 2$. This has important
implications, as the Gaussian hypothesis assumed by the Kalman filter theory is not satisfied. As a result, the analysis step can
yield an unrealistic posterior estimate, including negative concentrations.

This is illustrated in Fig. 2b, which shows the two-dimensional histogram plot for the posterior distributions resulting from
the LETKF. Clearly, the statistics of the analysis ensemble tend to become Gaussian and, as a result, the algorithm generates
an unrealistic ensemble which is not consistent with the non-Gaussian Bayes's theorem, introducing artificial negative values
for both ensemble mean and skewness.

In this work, we follow a simple approach to partially fix this problem by removing negative concentrations (a zero value
is assigned). This truncated state is no longer a solution of the original Kalman filter problem and the ability of this method
to produce an improved state should be explicitly demonstrated. The probability of obtaining nonphysical solutions increases
with the local radius for observations ($L_R$) and with the number of observations close to zero. For these reasons, global filters

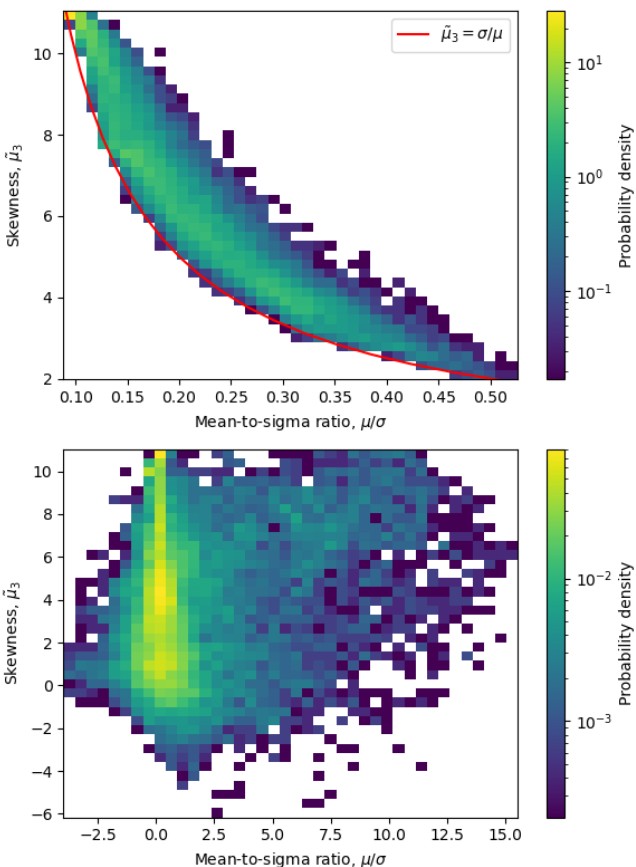

**Figure 2.** Two-dimensional histogram plot for the prior (top) and analysis (bottom) distributions showing the probability density for skewness ($\tilde{\mu}_3$) and mean-to-sigma ratio ($\mu/\sigma$) values, where $\mu$ refers to ensemble mean and $\sigma$ to standard deviation. Results correspond to the first assimilation cycle of the Raikoke experiment.

such as ETKF are not considered here. On the other hand, only observations with positive column mass exceeding a given threshold, related to the detection limit of satellite sensors, are assimilated.

In addition to removing negative data, we also explored an alternative definition of the vector state $x$ in terms of some nonlinear transformation $x = T(C)$, so that background concentration values close to zero are stretched out. A logarithmic function or the square root are two obvious options for $T$. In this way, the filtering process occurs in the transformed space and, after the analysis, concentration can be recovered by applying the inverse transformation, i.e., $C = T^{-1}(\overline{x})$. This "transformed state" approach failed with a logarithmic mapping due to the existence of few outliers leading to extremely large concentrations

when the inverse transformation was applied. In contrast, the square root transformation resulted in reasonable results and a stable filter. In practice, the square root transformation can be enabled by the user through the FALL3D input parameter SQRT_TRANSFORMATION, as indicated in Table 1.



## 2.4 Observation operator

The DA system supports assimilation of satellite-retrieved mass loading (i.e., the vertical column mass per unit area) of volcanic
ash and gases ($SO_2$ and $H_2O$). As a consequence, the objective is to reconstruct the three-dimensional concentration $C_i(x, y, z)$
field of each specie $i$ from a two-dimensional observational dataset. The observation operator $\mathbf{H}$, which projects a model state
$x \in \mathbb{R}^n$ onto the observation space, entails a vertical integration of concentration, a sum over different species (if multi-species
observations are being assimilated) and, finally, the interpolation to the observation coordinates. Note that, if the vector state
$x$ represents mass concentration, $\mathbf{H}$ is a linear operator. This is the main advantage of focusing on mass loading rather than on
other observable, e.g., aerosol optical depth, which would lead to a nonlinear observation operator.

The observation operator acting over the analysis vector defines a vector $y^a \in \mathbb{R}^p$ of analysed mass loading:

$$y^a = \mathbf{H}\overline{x}^a \tag{1}$$

where $\overline{x}^a$ is the assimilated state vector (analysis). See Sect. A for further details. In order to facilitate the visualisation and
enable a direct comparison with observations, the analysed mass loading, $y^a$, will be shown in the following figures. However,
if not explicitly stated otherwise, the full analysis state, i.e., $\overline{x}^a$, will be used to compute the evaluation metrics (see Sect. 2.7).

## 2.5 Ensemble generation

In order to generate a set of $m$ background states, FALL3D automatically perturbs Eruption Source Parameters (ESP) and
horizontal wind components from a reference value using either uniform or truncated normal distributions (Folch et al., 2021).
A Latin Hypercube Sampling (LHS, McKay et al., 1979) is used to efficiently sample the parameter space. Table 2 lists the
perturbed parameters in the twin and Raikoke DA experiments that are considered in this work (see Sect. 3).

## 2.6 Satellite retrievals

The satellite retrievals used for the Raikoke DA experiment are $SO_2$ mass loading retrievals derived from AHI/Himawari-8
measurements. Details of the retrieval method are described in Appendix B of Prata et al. (2021). The retrieval is based on the
strong absorption of $SO_2$ near the 7.3 μm wavelength and is generally only sensitive to upper-level ($\gtrsim 4$ km) $SO_2$ due to the
masking effect of water vapour absorption at lower levels in the atmosphere (Prata et al., 2004). A conservative estimate of the
relative uncertainty on these mass loading retrievals is 30%.

## 2.7 Evaluation metrics

When the true state $x_{tr} \in \mathbb{R}^n$ is known (e.g., experiments with synthetic observations as in Sect. 3.1), the difference between
the ensemble mean and the truth can be directly quantified using the average root-mean-square error:

$$\mathrm{RMSE} = \sqrt{\frac{\|\overline{x} - x_{tr}\|_2}{n}} \tag{2}$$





**Table 2.** Ensemble configuration for the twin and Raikoke experiments. In order to generate the ensemble, eruption source parameters (ESP) and wind components were perturbed around a reference value using either uniform or truncated normal distributions. The Latin Hypercube Sampling (LSH) method is used to sample the parameter space. The perturbed ESP are: eruption start time ($T_i$), source duration ($\Delta T$), eruption column height (H), mass emission rate (MER), parameters $A_s$ and $\lambda_s$ if the Suzuki vertical mass distribution, and top-hat thickness ($\Delta Z$).

| Parameter | Reference value | Distribution | Sampling range |
|---|---|---|---|
| *True state for Twin Experiment* | | | |
| H | 12-14 km[†] | - | - |
| Ash MER | Estimated[‡] | - | - |
| $A_s$ | 6 | - | - |
| $\lambda_s$ | 4 | - | - |
| $\Delta T$ | 6 h | - | - |
| U wind | WRF-ARW | - | - |
| V wind | WRF-ARW | - | - |
| *Ensemble for Twin Experiment* | | | |
| H | 10 km | Uniform | ±40% |
| Ash MER | $10^7$ kg/s | Fixed | - |
| $A_s$ | 6 | Gaussian | ±25% |
| $\lambda_s$ | 4 | Gaussian | ±25% |
| $\Delta T$ | 6 h | Fixed | - |
| U wind | WRF-ARW | Gaussian | ±25% |
| V wind | WRF-ARW | Gaussian | ±25% |
| *Ensemble for Raikoke Experiment* | | | |
| H | 12.5 km | Uniform | ±3 km |
| $SO_2$ MER | $2 \times 10^5$ kg/s | Uniform | ±20% |
| $\Delta Z$ | 2 km | Uniform | ±1 km |
| $T_i$ | 00 UTC[§] | Uniform | ±6 h |
| $\Delta T$ | 2 h | Uniform | ±1 h |
| U wind | GFS | Uniform | ±25% |
| V wind | GFS | Uniform | ±25% |

[†] Variable column height as in Fig. 3

[‡] Parameterisation from Degruyter and Bonadonna (2012)

[§] On 22 June 2019

In contrast, for the case involving real observations (see Sect. 3.2), the root-mean-square error is computed in the observation space according to:

$$\mathrm{RMSE}_o = \sqrt{\frac{\|\boldsymbol{y} - \boldsymbol{y}^a\|_2}{p}} \tag{3}$$





where $\boldsymbol{y} \in \mathbb{R}^p$ represents a vector with $p$ observations and $\boldsymbol{y}^a$ is the analysed mass loading vector defined by Eq. (1).

A measure of the uncertainty of the ensemble is given by the ensemble spread, $\sigma_e$. The domain-averaged spread can be defined in terms of the ensemble-based covariance matrix as:

$$\sigma_e = \sqrt{\frac{\mathrm{tr}(\mathbf{P_e})}{n}} \tag{4}$$

where $\mathbf{P_e}$ is the ensemble-based matrix for the covariance, defined by Eq. (A4) in Appendix. Note that the true state is not involved in this definition, meaning that this metric is independent of $\boldsymbol{x}_{tr}$.

Additionally, we consider also categorical metrics defined for model and observations from the exceedance (or not) of a given threshold. For example, in the case of categorical metrics for the total column mass loading, a true positive means that both model and observation exceed a given threshold value. The True Positive Rate or Probability of Detection (POD) is defined as the number of True Positives (TP) divided by the number of False Negatives (FN) plus True Positives (TP):

$$\mathrm{POD} = \frac{\mathrm{TP}}{\mathrm{FN+TP}} \tag{5}$$

The POD ranges from 0 to 1 (optimal) and, geometrically, it can be interpreted as the area of the intersection between the modelled and observed column mass contours, normalised by the area of the observation contour (Folch et al., 2021).

## 3 Numerical experiments

This section presents results from two numerical experiments aiming at evaluating the performance of the FALL3D+PDAF DA system under different filter configurations. The first experiment (twin experiment) is described in Sect. 3.1 and the second

experiment (Raikoke experiment) is described in Sect. 3.2. Table 3 summarises the model configuration defined for each experiment.

  A critical aspect in operational workflows is the computational cost required by the ensemble forecasting system. FALL3D has been proved to have a good strong scalability (above 90% of parallel efficiency) up to several thousands of processors (Folch et al., 2020). As the ensemble forecasting task is embarrassingly parallel, major constraints on computing time probably come

from the analysis step. Simulations were conducted on the Joliot-Curie supercomputer at the CEA's Very Large Computing Center (TGCC, France) using 1152 processors for the twin experiment (ensemble size: 48) and 3072 processors for the Raikoke experiment (ensemble size: 128). The typical computing times were of around 200 s (twin experiment) and 375 s (Raikoke experiment).

### 3.1 Twin experiment

Twin experiments are commonly used to evaluate DA methods. In this case, the *truth* state is generated by a model run in order to obtain a reference vector state. Synthetic observations are generated by adding random perturbations to the true state, which represent non-correlated observation errors. An ensemble forecast is then produced by perturbing a state estimate (different from the truth) and synthetic observations are assimilated. The performance of the ensemble filter can be evaluated by comparing the assimilation results with the true state.





**Table 3.** Model configuration parameters for the numerical experiments considered in this work.

| Parameter | Twin experiment | Raikoke experiment |
|---|---|---|
| Ensemble size | 48 | 128 |
| Grid size | $0.1° \times 0.1°$ | $0.2° \times 0.2°$ |
| Domain size | $195 \times 155 \times 50$ | $300 \times 150 \times 50$ |
| Species | 4 ash bins | $SO_2$ |
| TGSD | Estimated[†] | - |
| Run time | 36h | 72h |
| Emission source | Suzuki source[‡] | Top-hat source |
| Assimilation frequency | 3h | 3h |
| Assimilation start time | 6h | 18h |

[†] Costa et al. (2016a)

[‡] Pfeiffer et al. (2005)

The twin case study considers a fictitious eruption from Etna driven by WRF-ARW meteorological data in order to produce realistic atmospheric conditions. As stated in Table 3, a 36-h numerical simulation was performed considering an eruption lasting 6 h with a mass emission rate (MER) estimated from the eruptive column height ($H$) according to Degruyter and Bonadonna (2012). The (synthetic) time evolution of column height is shown in Fig. 3. In a previous study including several

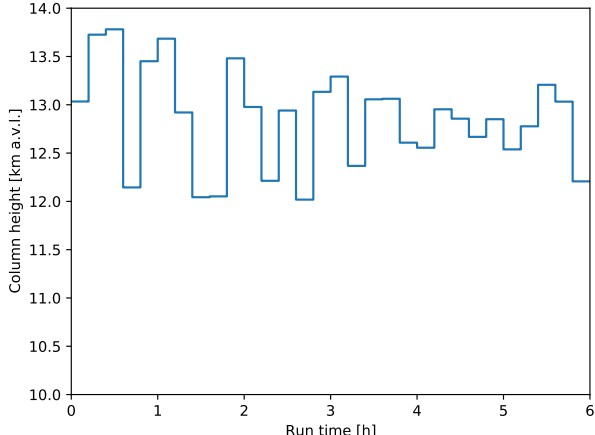

**Figure 3.** Time evolution of the eruption column height used to define the twin experiment true state. The 6-h duration eruption is characterised by multiple eruptive phases with duration of 20 minutes and column height randomly sampled within the range 12–14 km above the vent.

cases, Costa et al. (2016b) found a maximum column height variability of 30% for weak plumes and 10% for strong plumes.


Moreover, Suzuki et al. (2016a,b) showed that a variability of up to $\sim 20\%$ is common. Correspondingly, the twin experiment
in this work considers multiple eruptive phases with a duration of 20 minutes and a column height sampled from a uniform
probability distribution within the range 12–14 km above the vent (15.3–17.3 km above sea level). Such a time-varying source
term can result in complex cloud dynamics and represents a challenge for dispersion models and DA. The Suzuki plume option
was adopted for the vertical distribution of mass (Pfeiffer et al., 2005), and the total grain size distribution was estimated

from the time-varying column height following the parameterisation proposed by Costa et al. (2016a) assuming a magma
viscosity of $\eta = 10^5$ Pa-s. The computational domain has an horizontal resolution of $0.1°$ and a domain size of $n_x \times n_y \times n_z =$
$195 \times 155 \times 50$ grid cells. Simulations involve four fine ash bins ($n_b = 4$) with particle diameters $d < 10 \ \mu m$ and, consequently,
the dimension of the state vector $\boldsymbol{x}$ used in the assimilation cycle is $n = n_x \times n_y \times n_z \times n_b \approx 6 \times 10^6$.

Synthetic observations were generated by adding a Gaussian noise to the column mass loading computed from the true

state. As in Pardini et al. (2020), a conservative relative error of 40% is considered for both the synthetic observations and
the Raikoke $SO_2$ retrievals. In order to represent a realistic scenario where the range of valid measurements is restricted by
the instrumental detection limit, we assume mass loading observations are above a given threshold. For example, Prata and
Prata (2012) suggested a detection limit of $0.2 \ \mathrm{g\,m^{-2}}$, approximately, for SEVIRI retrievals of ash mass loading. On the other
hand, Mingari et al. (2020) found a good correlation between MODIS airborne ash detection products and the $0.1$-$\mathrm{g\,m^{-2}}$ mass

loading contours simulated by FALL3D. In this work, synthetic observations were defined assuming a mass loading threshold
of $0.15 \ \mathrm{g\,m^{-2}}$.

The twin experiment considers a 48-member ensemble and two types of simulations: (i) a free run without assimilation and,
(ii) a set of LETKF runs, where observations were incorporated with an assimilation frequency of 3 h beginning at $t = 6$ h
after the simulation start. To generate the ensemble, the column height was uniformly sampled around a reference value of

10 km with a perturbation range of 40% and assuming a fixed mass flow rate of $10^7 \ \mathrm{kg\,s^{-1}}$ (see Table 2). Since both eruption
column height and eruption rate are assumed to be constant here, no single member can actually reproduce the true state by
itself because the control run was defined from a time-varying source term (Fig. 3). Furthermore, the ensemble central column
height ($H = 10$ km) tends to underestimate the true column height. Consequently, the ensemble was not optimally constructed
to mimic a realistic situation in an operational forecasting workflow in which the exact column height is unknown.

### 3.1.1   Twin experiment results

The spatial distribution of mass loading is shown in Fig. 4 at simulation time $t = 18$ h after the eruption start time according
to the true state (4a), synthetic observations (4b), ensemble free run without assimilation (4c), and LETKF analysis (4d). In all
cases, the ash cloud for this idealised eruption is transported eastwards by upper-level winds. As expected, the free run case
shows a broader spatial distribution than the true state due to the ensemble spread. Moreover, the free run incorrectly predicts

the location of the column mass maximum occurring over the northern region of the cloud. In contrast, the analysed mass
loading field approaches the true state after a few assimilation cycles (Fig. 4d).

To quantify the impact of DA, the RMSE and ensemble spread were computed using Eqs. (2) and (4). Figure 5a shows the
time-averaged (over the whole simulated period) RMSE for different localisation radius and 2 multiplicative inflation factors



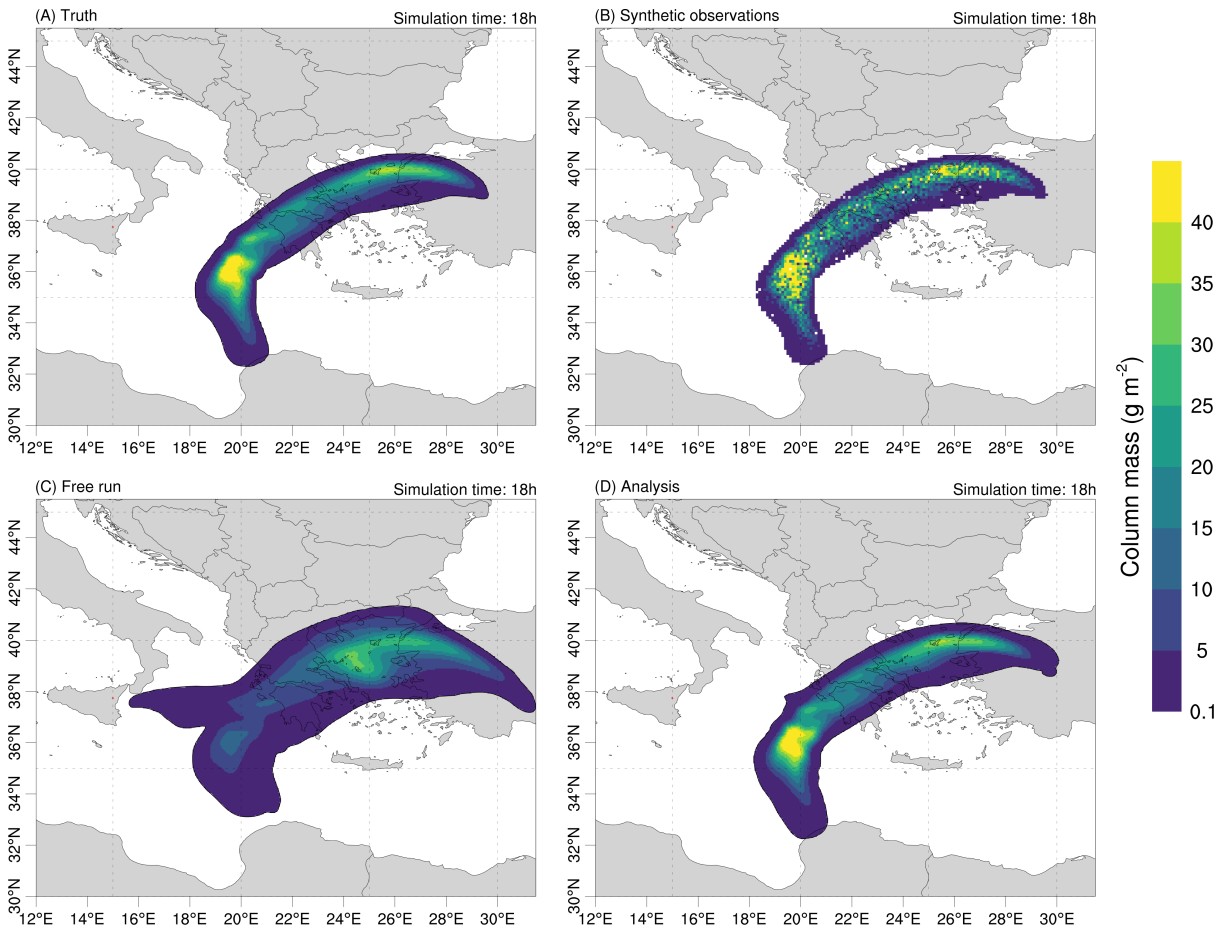

**Figure 4.** Spatial distribution of ash mass loading for the twin experiment at $t = 18$ h after eruption start. The true state (a) given by a single run assumes a time-varying emission. Synthetic observations (b) are generated from the truth by adding a Gaussian noise and assuming an observation error variance of 40%. The impact of the LTEKF DA becomes evident by comparing results from the free ensemble run without assimilation (c) with the analysed mass loading (d).

of $\lambda = 1$ (black triangles) and $\lambda = 1.2$ (red circles). Simulations were repeated three times to inspect the impact of the random

noise, and the resulting metrics averaged (solid lines). Despite the large scattered data, optimal localisation radius seems to be between $L_R = 2°$ and $L_R = 4°$ (20 to 40 grid cells), with a notorious degradation of performance for $L_R < 2°$. Increasing the inflation factor from $\lambda = 1.0$ to $\lambda = 1.2$ resulted in slightly smaller RMSE in most of the ensemble realisations (Fig. 5a). Hourly time series of the evaluation metrics are shown in Fig. 5b for the free and LETKF runs (analysis times are indicated by star symbols). The optimal parameters $L_R = 4°$ and $\lambda = 1.2$ were used here to configure the LETKF run. As expected for a

diffusive process without sources, the RMSE decreases from $t = 6$ h, when the eruption ends. Clearly, the LETKF simulation outperforms the free run. The impact of DA becomes more apparent by looking at the relative RMSE, i.e., the LETKF-to-free





ratio of RMSE. In the first assimilation cycle at $t = 6$ h, the relative RMSE decreases abruptly from 1 down to $\sim 0.2$. During successive assimilation cycles this ratio decreases further, suggesting that the analysis is converging to the true state.

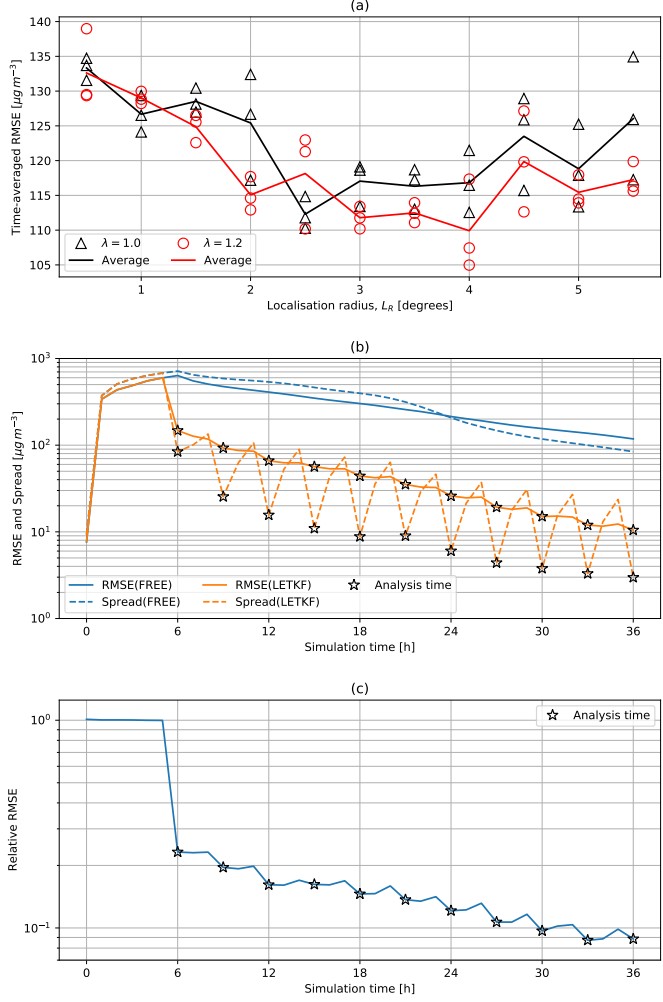

**Figure 5.** Evaluation metrics used for the twin experiment: (a) Time-averaged RMSE computed for different filter configurations and three ensemble realisations. Best performance was obtained for localisation radius, $L_R$, in the range 2–4° and an inflation factor of $\lambda = 1.2$; (b) Temporal evolution of ensemble spread and RMSE for the FREE and LETKF runs; and (c) Time series of LETKF-to-FREE ratio of RMSE. Assimilation times are stated by star symbols.

The ensemble spread should be close to the analysis error since under-dispersive ensembles are prone to filter divergence. As depicted in Fig. 5b, a steep decrease in spread occurs at each assimilation time, which is compensated by the ensemble variability introduced during each forecast period. The 3-h assimilation frequency turned out to be sufficient to keep spread just above the RMSE during each assimilation cycle, meaning that uncertainties are correctly represented by the ensemble.





In conclusion, the twin experiment shows that it is possible to reconstruct the original 3D model state of concentration field from an incomplete dataset of 2D measurements subject to uncertainty. A good filter performance was achieved despite the

fact that column mass data below $0.15 \, \mathrm{g \, m^{-2}}$ was discarded, i.e., that only a fraction of the available column mass data was actually assimilated.

## 3.2    The 2019 Raikoke eruption

On 21 June 2019, the Raikoke volcano ($48.292°$ N, $153.25°$ E) in the Kuril Islands (Russia) had a significant eruption that disrupted major flight routes across the North Pacific (Prata et al., 2021). The eruption injected ash and gases into the atmosphere

in a sequence of around 10 eruptive pulses, from the initial explosive phase at 18:00 UTC on 21 June until 10:00 UTC on 22 June (Muser et al., 2020). The eruption sequence was captured by the Himawari-8 satellite at both IR and visible wavelengths. A remarkable amount of $SO_2$ was injected into the atmosphere during these explosive phases, producing a long-range transport of $SO_2$ that could be detected by satellite instrumentation.

In order to simulate this event, the FALL3D computational domain was configured using an horizontal resolution of $0.2°$

and a domain size of $n_x \times n_y \times n_z = 300 \times 150 \times 50$ grid cells. In this case, the state vector $\boldsymbol{x}$ includes only $SO_2$ and has a size of $n \approx 2 \times 10^6$. For this experiment, 72-h numerical simulations were conducted starting on 21 June 2019 at 18:00 UTC using 128 ensemble members. A free run without DA and several LETKF runs were performed for comparative purposes. Assimilation starts on 22 June 2019 at 12:00 UTC with a frequency of 3 h for the successive assimilation cycles. The top-hat option was adopted for the vertical mass distribution in the source term, i.e., the source term is defined by a uniform

mass distribution along a layer of thickness $\Delta Z$ and top at height $H$. Both parameters were perturbed with central values of $\Delta Z = 2$ km and $H = 12.5$ km above sea level. In addition, mass emission rate (MER), start time and duration of eruption, and wind components were also perturbed. Specifically, the emission start time was uniformly sampled between 18:00 UTC on 21 June and 06:00 UTC on 22 June, assuming a duration of $\Delta T = 2 \pm 1$ h for each ensemble member. Note that the eruption total time for Raikoke was around of 14 h, meaning that each ensemble member represents a possible eruptive phase lasting

a fraction of the total eruption time. This approach was adopted in order to reproduce a multi-phase eruptive scenario with a complex time-varying emission source term. In this case, the real state involves a mixture of multiple ensemble members with weights to be determined by the analysis step.

The list of model parameters used to generate the ensemble are detailed in Table 2 and Table 3 summarises the general model configuration used in the Raikoke experiment. The dispersal model was driven by meteorological data from the Global

Forecast System (GFS) model instead of using reanalysis data in order to replicate an operational forecasting environment.

### 3.2.1    Raikoke experiment results

Figure 6 compares the spatial distribution of $SO_2$ mass loading according to the satellite retrievals (left panel), free run (central panel), and analysis (right panel) at three time instants. On 22 June, the volcanic plume is influenced by upper-level zonal winds and moves eastwards crossing the 180[th] meridian. From 23 June, the plume of sulphur dioxide gets trapped within the







**Figure 6.** Spatial distribution of SO$_2$ mass loading at different time instants: (A-C) 22 June at 15:00 UTC, (D-F) 23 June at 00:00 UTC, and (G-I) 23 June at 09:00 UTC. The column panels show: observations (left panel), free run (central panel), and analysed mass loading (right panel).

cyclonic circulation of the Aleutian low causing the airborne material to spiral counterclockwise for several days (Kloss et al., 2021).

In order to assess the filter performance, two quantitative metrics defined in Sect. 2.7 will be considered below. First, the root-mean-square error (RMSE$_o$) is computed in the observation space using Eq. (3). Figure 7a shows the RMSE$_o$ for all the analysis




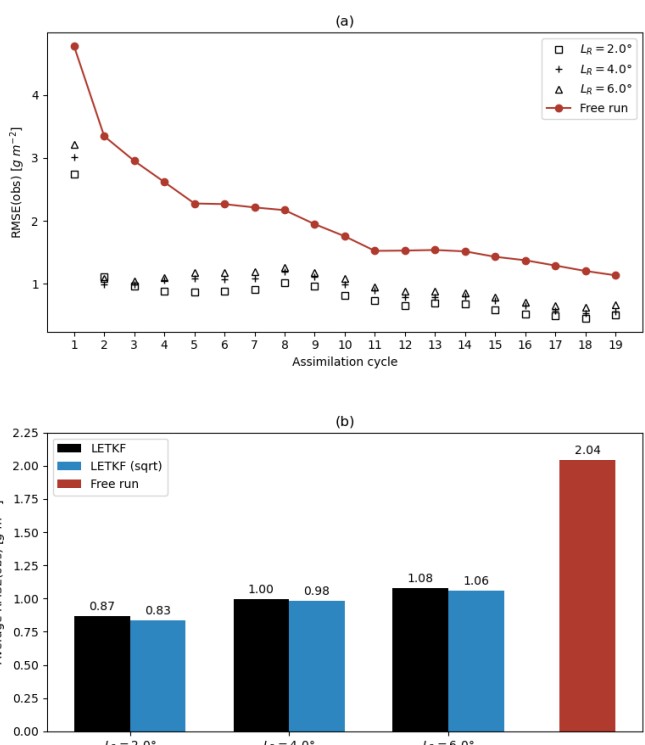

**Figure 7.** $RMSE_o$ computed in the observation space for the Raikoke experiment: (a) time series of $RMSE_o$ at analysis times for the free (red line) and LETKF runs for different localisation radius ($L_R = 2°$, $L_R = 4°$ and $L_R = 6°$) and (b) same with time-averaged values.

states using different localisation radius ($L_R = 2°$, $4°$ and $6°$). Despite the occurrence of nonphysical solutions (grid cells with
negative concentrations) during the first assimilation cycle, the truncated LETKF solutions outperform the free run in all cases. After successive assimilation cycles, the ensemble analysis becomes closer to a Gaussian distribution and the probability of obtaining nonphysical solutions diminishes. Results in Fig. 7 also show that $RMSE_o$ decreases with the localisation radius. Specifically, the time-averaged $RMSE_o$ (Fig. 7b) decreased from $1.08 \, \mathrm{g \, m^{-2}}$ ($L_R = 6°$) to $0.87 \, \mathrm{g \, m^{-2}}$ ($L_R = 2°$). Overall, the analysis errors were decreased by more than 50% relative to the free run errors. However, it is important to highlight that it is
not possible to infer the filter performance was improved by decreasing the localisation radius as no true state is now available to compute the actual RMSE (see Sect. 3.1). Finally, Fig. 7b also shows results for the LETKF(sqrt) simulations, where the option `SQRT_TRANSFORMATION` was enabled (see Sect. 2.3), meaning the vector state was constructed from the square root of the concentration. This approach resulted in a slightly smaller $RMSE_o$, but the impact does not appear to be significant.

While the free run results show a very poor correlation between observed and modelled $SO_2$ mass loading, a clear correlation
emerges after a few assimilation cycles in the LETKF simulations. As an example, Fig. 8 shows a comparison between the observed and analysed mass loading at the fourth assimilation cycle. A systematic bias, likely caused by the characteristics





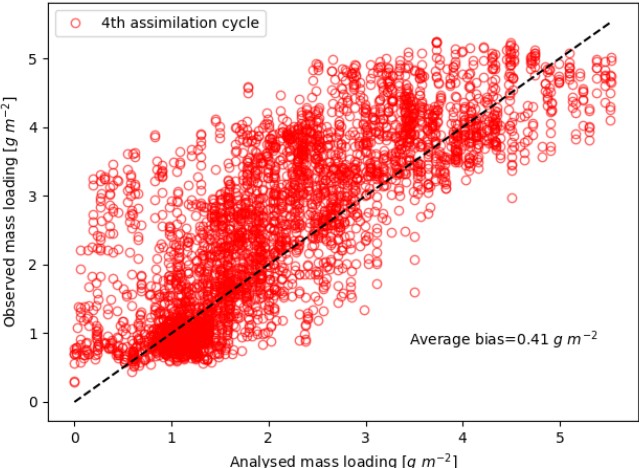

**Figure 8.** Comparison of SO$_2$ mass loading observations and the analysis state at the fourth assimilation cycle. In general, analysis underestimates observations. In this case, an average bias of $0.41\ \mathrm{g\,m^{-2}}$ was found.

of the ensemble distribution, was found at each assimilation cycle and analysis tends to underestimate observations. In this particular cycle, for instance, an average bias of $0.41\ \mathrm{g\,m^{-2}}$ was found.

The spatial distribution of observed and analysed mass loading for the SO$_2$ cloud on 23 June at 12:00 UTC are shown in
Fig. 9 along with the cloud top height derived from the analysed state. A complete sequence of the temporal evolution for this figure can be found in the supplementary material. The cloud top height is defined as the upper height of a given iso-concentration contour ($50\ \mu g\ m^{-3}$ was assumed here). The model can correctly capture the position of the SO$_2$ plume for mass loading contours above $1\ \mathrm{g\,m^{-2}}$. However, no observations are available below $0.5\ \mathrm{g\,m^{-2}}$ (e.g., see Fig. 8), making challenging the comparison for low mass loading values. Specifically, the analysed mass loading indicates the existence of a
low-level cloud in the southern region that could not be detected by the satellite retrievals.

Finally, we computed categorical metrics based on the $1$-$\mathrm{g\,m^{-2}}$ contour of SO$_2$ mass loading. The resulting maps are shown in Fig. 10 for three time instants. The complete time sequence can be found in the supplementary material. The plume dynamics according to the free run (top panel) follows a similar pattern to that found in previous simulations by Prata et al. (2021). Specifically, the free run (solid red line) and observed contours (green shaded area) diverge after a few time steps. In contrast,
the contours corresponding to simulations with data assimilation evolve concurrently with observations for all simulated times (Fig. 10, bottom panel). The performance of the simulations can be quantified through the POD categorical metric (Eq. 5), which can be interpreted geometrically as the ratio between the intersection area delimited by both the observation and model contours and the total area of the observation contour. Figure 11 shows the temporal evolution of the POD. After a forecast time of around $t = 18$h, this metrics tends to decrease monotonically for the free run, whereas it remains close to the optimal value
(POD=1) along all time steps when data assimilation is enabled. A sudden increase on this metrics occurs at each assimilation


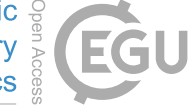


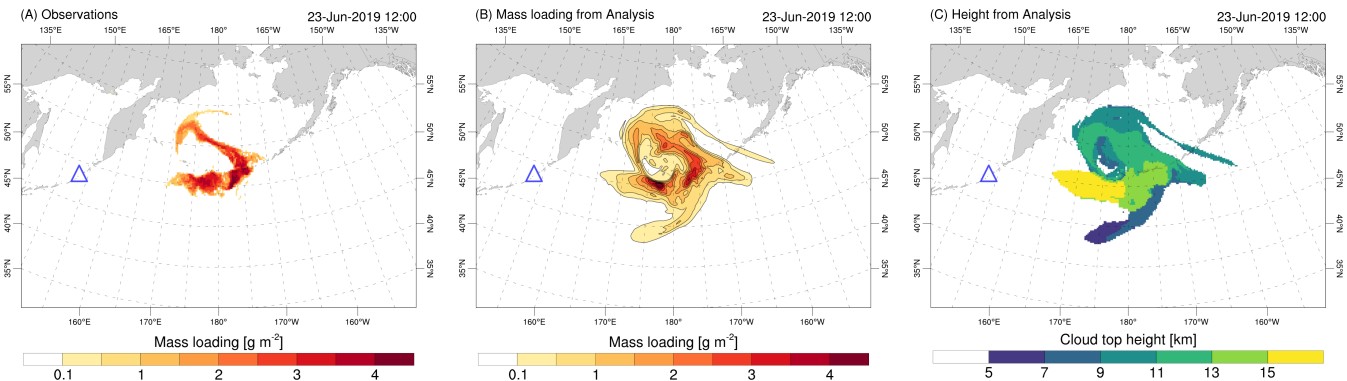

**Figure 9.** SO$_2$ cloud for the 2019 Raikoke eruption on 23 June at 12:00 UTC. Observed (a) and modelled (b) mass loading are compared at the same instant of time. In addition, the cloud top height (c) derived from the analysed state is also shown. The LETKF run was configured assuming a localisation radius of $L_R = 2°$ and an assimilation frequency of 3h, starting on 22 June 2019 at 12:00 UTC.

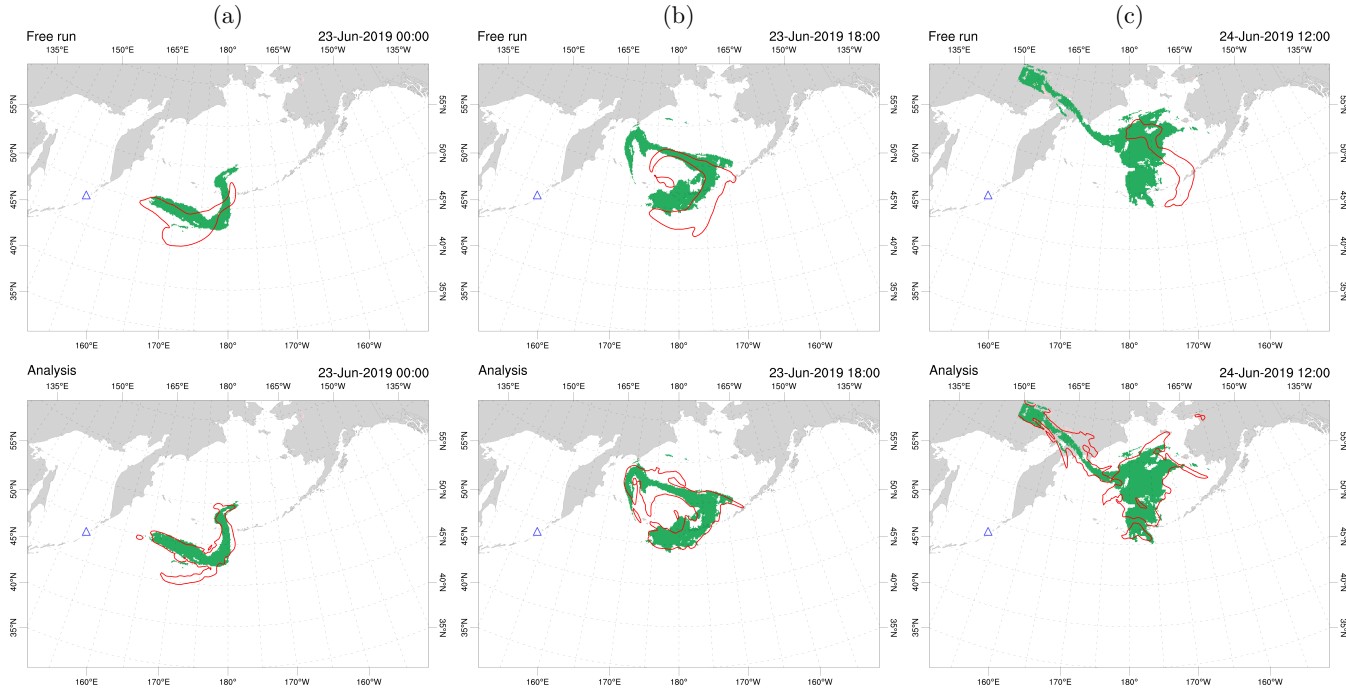

**Figure 10.** Maps of $1\,\mathrm{g\,m^{-2}}$ contour of SO$_2$ mass loading according to observations (green shaded area) and model (solid red line) for different instants of time: (a) 00:00 UTC on 23 June 2019, (b) 18:00 UTC on 23 June 2019, and (c) 12:00 UTC on 24 June 2019. Model results corresponding to the free run (top panel) and the analysis (bottom panel) are compared. A localisation radius of $L_R = 2°$ was defined for the data assimilation method.





cycle (square symbols), clearly visible from Fig. 11, which prevents this metrics from degrading significantly. In conclusion, POD remains in the range around 0.8–0.9.

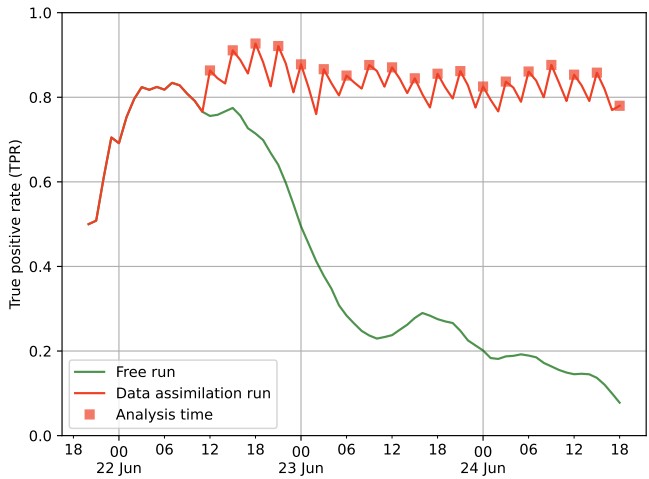

**Figure 11.** Temporal evolution of the Probability of Detection (POD) metrics according to the free and data assimilation runs for the Raikoke experiment. A localisation radius of $L_R = 2°$ was defined for the data assimilation simulation.

## 4 Discussion

In this work, a localised version of the ensemble Kalman filter LETKF has shown to be a promising alternative for assimi-
lation of volcanic aerosols. Despite the limitations of this method, resulting in suboptimal filter performance, our findings do
nevertheless show that a significant improvement of evaluation metrics was achieved.

Ensemble Kalman Filters give an optimal state estimate under the following implicit assumptions: (i) the distribution of
the background is Gaussian, (ii) the observational error has Gaussian distribution, and (iii) the forward model and observation
operator are linear. FALL3D is a dispersal model with weak nonlinear terms and modelling complex multi-phase eruptions
entails the contribution of multiple ensemble members to properly represent the model state. However, in this work it has been
shown that skewness is a significant issue and the condition (i) is largely violated resulting in suboptimal behavior from the
EnKF.

Different approaches have been proposed for dealing with non-Gaussianity, including variable transformations (e.g., Zhou
et al., 2011; Amezcua and Van Leeuwen, 2014) and Bayesian approaches, such as particle filter (e.g., van Leeuwen and Ades,
2013) or the nonlinear ensemble transform filter (NETF, Tödter and Ahrens, 2015). Unfortunately, these methods suffer from
a series of pitfalls. For instance, variable transformation applied to skewed prior distributions would require highly nonlinear
transformations to obtain state variables fulfilling the Gaussianity conditions, again leading to suboptimal states. On the other
hand, particle filters and NETF are exposed to weight collapse due to the so-called curse of dimensionality which would result





in a poor performance in complex eruptive scenarios with stochastic time-varying emission source parameters. In addition,
although the aforementioned methods may be suitable for problems involving highly nonlinear processes, ensemble-based
Kalman filters are expected to work better for linear and Gaussian problems (e.g., Tödter and Ahrens, 2015). In consequence,
it is not clear which of these methods would result in a better performance for linear (or weakly nonlinear) and non-Gaussian
problems.

Promising results obtained in this work using LETKF suggest that the natural approach for dealing with assimilation of vol-
canic aerosols in future research should focus on ensemble Kalman filters in which the Gaussian assumption is not made at all.
For example, Bishop (2016) proposed an ensemble Kalman filtering for highly skewed non-negative uncertainty distributions.
This approach allows the EnKF to be generalised with few coding changes and little additional computational expense. Finally,
higher-order EnKFs have also been proposed (e.g., Hodyss, 2012; Hodyss and Campbell, 2013) and could potentially address
the aforementioned issues.

## 5   Conclusions

A detailed study has been conducted in order to assess the feasibility of using ensemble-based Kalman filters for data assim-
ilation (DA) of volcanic aerosols. To this purpose, a new DA system based on coupling the FALL3D dispersal model with
the Parallel Data Assimilation Framework (PDAF) has been implemented in the latest release of FALL3D (v8.2). The system
supports online-coupled DA, can be run in parallel exploiting high-performance computing (HPC) resources, and is suitable for
an operational workflow. The computing time required by the numerical simulations carried out in this work ranges between 2
(twin experiment) and 6 (Raikoke experiment) minutes.

One of the major assumptions in the (ensemble) Kalman filters is that the prior model errors and the observation noise are
Gaussian. However, ensemble forecasts of volcanic aerosols yield to a non-Gaussian prior, with positively skewed distributions.
Consequently, the ability of the assimilation technique to produce an improved model state compatible with the available
observations require explicit verification.

We carried out two numerical experiments in which mass loading data was assimilated using the Local Ensemble Transform
Kalman Filter (LETKF). Both test cases are characterised by a complex plume dynamics and time-varying eruptive source pa-
rameters (ESP) that pose a challenge to dispersion models, especially in operational environments. The complexities involved
in the definition of the source term were intentionally discarded in the prior ensemble construction in order to replicate an
operational environment, where such variations are typically unknown. A constant eruption column height (H) and mass emis-
sion rate (MER) were assumed for each ensemble member, meaning that no member can individually reproduce the case study
correctly. In the twin experiment, the analysis converged to the true state if observations are continuously assimilated with
a frequency of 3 h. In the second experiment, involving the $SO_2$ plume produced by the 2019 Raikoke eruption, categorical
metrics (POD) also remain close to optimal values as long as observations are continuously assimilated every 3 h.





Even though the results presented here are encouraging, the proposed truncated LETKF methodology is not optimal and should be tested in broader contexts and under different scenarios. We also encourage the community to test and develop more appropriate methodologies for positively skewed, non-Gaussian prior distributions.

*Code availability.*   FALL3D-8.2 is available under the version 3 of the GNU General Public License (GPL) at https://gitlab.com/fall3d-distribution (last access: 11 November 2020). The PDAF code (version 1.14 was used here) and full documentation are available at http://pdaf.awi.de
(last access: 11 November 2020).

## Appendix A:  Theoretical framework: ensemble Kalman filters

In sequential schemes, the assimilation process is characterised by a sequence of steps involving a forecast step and a subsequent analysis in which the *a posteriori* estimate is obtained from the *a priori* forecast state by incorporating observational information. The Kalman Filter (KF) is a sequential DA method that provides an optimal solution for linear models and linear
observation operators under certain assumptions (Kalman, 1960). If the background state of a physical system is represented by a single vector $\overline{x}^b$ of size $n$ and error covariance matrix $\mathbf{P}^b \in \mathbb{R}^{n \times n}$, the analysis step of the KF consists on determining an analysis state estimate $\overline{x}^a$ and its associated covariance matrix $\mathbf{P}^a$ given a vector of observations $y \in \mathbb{R}^p$. In addition to linearity, the KF also assumes Gaussian distributions for model errors and observation noise. As a result, the multivariate Gaussian prior density function is described by two moments, i.e., $\overline{x}^b$ and $\mathbf{P}^b$ (e.g., see Tödter and Ahrens, 2015). In the analysis step,
the moments are updated according to the KF equations:

$$\overline{x}^a = \overline{x}^b + \mathbf{K}(y - \mathbf{H}\overline{x}^b) \tag{A1a}$$
$$\mathbf{P}^a = \mathbf{P}^b - \mathbf{K}\mathbf{H}\mathbf{P}^b \tag{A1b}$$

where $\mathbf{H} \in \mathbb{R}^{p \times n}$ is the observation operator that translates a model state $x$ into the observation space, and $\mathbf{K} \in \mathbb{R}^{n \times p}$ is the so-called Kalman gain matrix given by:

$$\mathbf{K} = \mathbf{P}^b \mathbf{H}^\intercal (\mathbf{H} \mathbf{P}^b \mathbf{H}^\intercal + \mathbf{R})^{-1} \tag{A2}$$

where $\mathbf{R} \in \mathbb{R}^{p \times p}$ is the observation error covariance matrix. Note that any reference to time indices is omitted and the $b$ and $a$ superindexes refer to the background or *a priori* state and to the analysis or *a posteriori* state, respectively. The Kalman gain matrix assigns relative weights to observations. A high-gain filtering implies more weight to measurements whereas a low-gain filtering tends to follow the model more closely.
The Ensemble Kalman Filter (EnKF) is a family of methods in which the state estimate of the system is represented by an ensemble of system states that actually provide a Monte Carlo approximation of the KF and replace the original covariance matrix by a sample covariance matrix $\mathbf{P}_e$ computed from the ensemble (Evensen, 1994). One of the most important practical advantages of ensemble-based techniques is the independence of the filter algorithm on the specific forward model. The




background ensemble state represents therefore the best estimate of the prior probability density function. Given an ensemble
of size $m$, consisting of $m$ model realisations (ensemble members) characterised by the vectors $\boldsymbol{x}_i$ $(i = 1, \ldots, m)$ at a certain
time, the state estimate in the EnKF is given by the ensemble mean

$$\overline{\boldsymbol{x}} = \frac{1}{m} \sum_{i=1}^{m} \boldsymbol{x}_i \tag{A3}$$

and the ensemble-based covariance matrix $\mathbf{P}_e \in \mathbb{R}^{n \times n}$ can be expressed in terms of the matrix of ensemble perturbations
$\mathbf{X} \in \mathbb{R}^{n \times m}$ as

$$\mathbf{P}_e = \mathbf{X}\mathbf{X}^{\mathsf{T}} \tag{A4}$$

where $\mathbf{X}$ is defined by

$$\mathbf{X} = \frac{1}{\sqrt{m-1}} [\boldsymbol{x}_1 - \overline{\boldsymbol{x}}, \ldots, \boldsymbol{x}_m - \overline{\boldsymbol{x}}] \tag{A5}$$

Given a background state $\{\boldsymbol{x}_i^b : i = 1, 2, \ldots, m\}$ and a set of observations represented by the vector $\boldsymbol{y} \in \mathbb{R}^p$, the analysis
step consists on determining an ensemble $\{\boldsymbol{x}_i^a : i = 1, 2, \ldots, m\}$ in agreement with Eqs. (A1a,b) but formulated in terms of
the ensemble-based mean and covariance matrices. The ensemble mean is updated using the standard KF analysis Eq. (A1a),
expressed in terms of the ensemble-based matrices for the covariance and for the Kalman gain:

$$\overline{\boldsymbol{x}}^a = \overline{\boldsymbol{x}}^b + \mathbf{K}_e (\boldsymbol{y} - \mathbf{H}\overline{\boldsymbol{x}}^b) \tag{A6}$$

where

$$\mathbf{K}_e = \mathbf{X}^b \mathbf{Y}^{\mathsf{T}} (\mathbf{Y}\mathbf{Y}^{\mathsf{T}} + \mathbf{R})^{-1} \tag{A7}$$

where we defined $\mathbf{Y} = \mathbf{H}\mathbf{X}^b$. Note that, with respect to Eq. (A2), the ensemble-based gain matrix $\mathbf{K}_e$ considers the ensemble
background perturbations $\mathbf{X}^b$ and their projections onto the observation space through the matrix $\mathbf{Y} \in \mathbb{R}^{p \times m}$. In this way, the
best estimate of the current state is determined in the analysis step through a weighted linear combination of the prior ensemble
perturbations.

Different EnFK methods vary depending on how the ensemble analysis is defined so that the update for the ensemble
covariance matrix given by Eq. (A4) is consistent with the original KF formulation (A1b). Most formulations can be divided
into two major categories, the *stochastic* (e.g., the perturbed observations-based EnKF formulation from Burgers et al., 1998)
and the *deterministic* approaches (Houtekamer and Zhang, 2016). The latter group includes the so-called square-root filters
that uses deterministic algorithms to generate the analysis ensemble (Nerger et al., 2012).

The Ensemble Transform Kalman Filter (ETKF, Bishop et al., 2001) is a popular square-root filter formulation that will be
considered in this work. A square-root filter requires a matrix $\mathbf{W} \in \mathbb{R}^{m \times m}$ to transform the ensemble perturbations according
to:

$$\mathbf{X}^a = \mathbf{X}^b \mathbf{W} \tag{A8}$$





In order to obtain the ensemble perturbations, the covariance update is required to be consistent with the original KF formulation given by Eq. (A1b), leading to (e.g., see Carrassi et al., 2018):

$$(\mathbf{X}^a)(\mathbf{X}^a)^\intercal = (\mathbf{X}^b)\mathbf{A}(\mathbf{X}^b)^\intercal \tag{A9}$$

where $\mathbf{A} \in \mathbb{R}^{m \times m}$ is the so-called *transform matrix*, defined by $\mathbf{A}^{-1} = \mathbb{1} + \mathbf{Y}^\intercal \mathbf{R}^{-1} \mathbf{Y}$ (Nerger et al., 2012). If the square root is denoted by $\mathbf{C}$ (i.e., $\mathbf{C}\mathbf{C}^\intercal = \mathbf{A}$), the weight matrix $\mathbf{W}$ is assumed to be expressed as:

$$\mathbf{W} = \mathbf{C}\mathbf{\Lambda} \tag{A10}$$

where $\mathbf{\Lambda} \in \mathbf{R}^{m \times m}$ is any orthogonal matrix preserving the ensemble mean (see Sect. 2.1). In this work, the symmetric square
root is used to define $\mathbf{C}$ according to the symmetric factorisation

$$\mathbf{C} = \mathbf{U}\mathbf{S}^{-1/2}\mathbf{U}^\intercal \tag{A11}$$

using the singular value decomposition: $\mathbf{U}\mathbf{S}\mathbf{V} = \mathbf{A}^{-1}$. This definition of the root square matrix ensures that the ensemble mean is preserved (Hunt et al., 2007).

The application of ensemble filters in geophysical systems can lead to spurious correlations and underestimations of the
ensemble spread due to a limited size of the ensemble, sampling errors, and model errors (Anderson and Anderson, 1999). The problem of variance underestimation (filter collapse) is usually addressed by using inflation methods, whereas localisation is adopted to suppress spurious correlations. In particular, we consider a multiplicative factor $\lambda > 1$ to inflate the covariance matrix $\mathbf{P_e} \rightarrow \lambda^2 \mathbf{P_e}$, which is equivalent to multiplying $\mathbf{X}^b$ by $\lambda$. This inflation-controlling parameter has to be experimentally tuned.

To address the problem of spurious correlations, Hunt et al. (2007) proposed the localised version of the ETKF (i.e., LETKF). A step-by-step procedure to implement the LETKF algorithm can be found in Hunt et al. (2007). Both ETKF and LETKF represent practical methods for data assimilation suitable for high-dimensional systems, relatively easy to implement, computationally efficient, and have been implemented in the FALL3D+PDAF system.

*Author contributions.* Conceptualisation, L.M.; Methodology, L.M., A.F., A.T.P., F.P., G.M., A.C.; Software, A.F., L.M.; Resources, A.T.P.;
Writing—original draft, L.M.; Writing—review and editing, L.M., A.F., A.T.P., F.P., G.M., A.C.; Visualisation, L.M.; Supervision, A.F.; Funding Acquisition, A.F. All authors have read and approved the final version of the manuscript.

*Competing interests.* The authors declare that they have no conflict of interest.

*Acknowledgements.* This work has been partially funded by the H2020 Center of Excellence for Exascale in Solid Earth (ChEESE) under the Grant Agreement No. 823844. A.T.P. acknowledges funding from the the European Commission, H2020 Marie Skłodowska-Curie Actions



(STARS (grant no. 754433)). We also acknowledge the Partnership for Advanced Computing in Europe (PRACE) for awarding us access to Joliot-Curie supercomputer at the CEA's Very Large Computing Center (TGCC, France).



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
