# Peer review of "Data Assimilation of Volcanic Aerosols using FALL3D+PDAF"

_Atmospheric Chemistry and Physics, 2021_

## Referee Comment (RC1)

Overview: The authors apply an ensemble data assimilation method (LETKF) to obtain optimal estimates of volcanic ash/SO2 concentrations using the FALL3D dispersion model. For ash simulations, synthetic data are used while for SO2 simulations, satellite retrievals obtained during the 2019 eruption of Raikoke are used. Both experiments yielded significantly better results than reference experiments in which no data assimilation was employed. However, it was noted that during the first assimilation cycle in which the prior ensemble was based on straightforward sampling of model parameter uncertainty ranges, the probability distribution was non-Gaussian, resulting in unphysical (negative) optimal concentration values in the posterior ensemble. This was attributed to the Gaussian assumption underpinning the LETKF.

General comments:

This work is scientifically sound and will be of interest to the volcanic ash dispersion modelling community. I recommend that the authors consider the following issues in the manuscript prior to publication, which are mostly related to the presentation.

1. More clarity is needed about how the details of the algorithm. In my view Figure 1 is nice but doesn't really help the reader understand what is actually being done. For example, even after the reading the whole paper it was not 100% clear to me how the ensembles were generated at each cycle. Do you initialize an ensemble of dispersion models using prior uncertainty estimates at t0 and compute analysis at t1, then use analysis at t1 to re-initialize the dispersion model and propagate to t2 and so on? That would mean that the initialization at the first step (volcanic source?) is quite different from initialization at subsequent steps (distal?). Please provide more concrete details so the reader doesn't need to guess.

2. On reading, it feels like the appendix was originally part of an earlier chapter. I would suggest that the authors either perhaps shorten the appendix and then insert it at an earlier stage as part of the methodology section or make an effort to make sure that discussion in Section 2 is self-contained and does not require the reader to read the appendix first. Some of the specific comments below are related to this issue.

Specific Comments:

Introduction: There is a substantial body work on 'inverse modelling' methods using satellite retrievals of volcanic ash that has not been mentioned. See for example list of citations in Zidikheri, Meelis J., and Chris Lucas. "Improving Ensemble Volcanic Ash Forecasts by Direct Insertion of Satellite Data and Ensemble Filtering." *Atmosphere* 12.9 (2021): 1215. It would also be useful to mention what the DA method in this manuscript can do that these other approaches cannot do given that these methods also use observations to improve the forecasts.

Lines 100-105: "Background error covariance" is mentioned in Line 105 but it wouldn't be clear to readers unfamiliar with DA methods what the word "background" is referring to. It would be helpful to define "background forecast" ( = " a priori forecast") earlier in the paragraph. Might also be useful to mention why the error covariance is important in DA.

Line 128: Last sentence of paragraph is hard to understand. What do you mean "filter operations are performed exclusively by ranks…". What "ranks"?

Line 134: Sentence stating that LETKF is "more realistic for volcanic ash" than ETKF might need a reference (or explain why you think this would be the case). Also, this statement is rather puzzling given that you state in the abstract that LETKF didn't work very well. I think a summary of the

differences between ETKF and LETKF might be needed here – including a brief discussion of the need for localisation in ensemble DA methods in general. Many readers will probably not have the time or inclination to read the appendix in detail even those details are available there. See also General Comment #2.

Line 146: "range" – is this is the localisation radius? "inflation factor" – needs explanation.

Lines 164-166: I didn't really understand this explanation for why the ensemble forecast prior PDF forecast would be skewed. Isn't the skewness just a consequence of the way the prior ensemble is constructed? Could not in principle the prior uncertainty be sampled in such a way so as to yield a more symmetric distribution?

Line 203: the overbar needs explanation (ensemble mean?)

Lines 212-216: Is there a reason for focussing on SO2 rather than volcanic ash retrievals here? Ash concentrations (rather than SO2) are of more interest in practical applications.

---

## Referee Comment (RC2)

**Review of "Data Assimilation of Volcanic Aerosols using FALL3D+PDAF"**

Leonardo Mingari, Arnau Folch, Andrew T. Prata, Federica Pardini4, Giovanni Macedonio, and Antonio Costa

**General comments:**

This paper presents the implementation of the coupling between the FALL-3D dispersion model and the Parallel Data Assimilation Framework to create an ensemble data assimilation method suitable for the assimilation of volcanic ash and sulphur dioxide. The new system is applied to two scenarios:

1. A hypothetical eruption of Etna with synthetic volcanic ash satellite retrievals assimilated
2. The assimilation of sulphur dioxide retrievals following the 2019 eruption of Raikoke

Although the initial implementation of the Local Ensemble Kalman Filter may lead to non-physical solutions, the authors show that a truncated version of this filter leads to a dramatic improvement in the location of ash/SO2 compared to a free running simulation without assimilation.

The paper is generally well written, scientifically sound and interesting. I recommend that is published subject to some minor revisions that I have outlined below.

**Specific comments:**

The introduction could benefit from the addition of a discussion of inversion modelling that can be used to constrain emission rates and plume height. Also, there is no mention of data insertion which is the simplest form of data assimilation.

The authors motivate the study by citing the impacts of volcanic ash on aviation, but the second set of experiments focus on the assimilation of sulphur dioxide. What was the reason for this? There are satellite retrievals of ash available for this eruption or are they too patchy? Is the fact that the satellite can only "see" the distal ash plume a problem?

I am unsure how Figure 1 enhances the readers understanding of the method – how is the ensemble constructed? Also, does the assimilation of the satellite retrievals take into account their uncertainty?

L150 You refer to something in the appendix – does this part of the appendix to be worked into the main body of the text?

The authors show that the prior pdf associated with the ensemble forecast tends to be skewed possibly leading to the unrealistic posterior estimate as the Gaussian assumption in Kalman filter theory is not satisfied. Can the prior pdf be modified by different parameter sampling strategy or constructing the prior in a different way?

Figure 6 and the comparison of observations, free run and analysis of the Raikoke eruption – are the distributions shown for the free run and analysis the ensemble means? In panel G, is the southern branch of ash missing due to the presence of meteorological cloud?

Figure 9 I really like panel C as you can see the ascent in the cyclone. It would be nice to see a similar plot for the free running ensemble. This might help explain the large differences between the free running simulation and observations seen in Figure 10. Can FALL-3D represent diabatic heating which can also cause ascent?

Assimilation is expensive - can anything be gained/lost from more/less frequent assimilation?

**Technical corrections:**

L2 and 25 Unsure what is meant here by "infrastructures"

L67 Change "enabled to quantify model uncertainties" to "enabled the quantification of model uncertainties"

L123 Unsure what is meant here by "embarrassingly (or perfectly)". The same on L244.

L136 Is there a reference for the "realistic results" you mention?

L146 Is the local range referred to here the same as $L_R$? Can you expand on the inflation factor that is referred to?

Table 2 caption last line – change if to of

Table 2 WRF-ARW needs to be defined

Equation 4 What does tr mean? Is n the number of ensemble members?

Table 3 Change grid size to resolution, Domain size to number of grid points, expand TGSD

Figure 3 This seems to be a very complex column height profile. Is it representative of what might be used in operations?

L279 It would be nice to remind the reader that it is a 36-hour forecast being performed here

L280 Why is the flow rate fixed? Could this also be perturbed or determined from the perturbed plume height?

L296 Is there a reference for the "notorious degradation"?

L326 What was the motivation for using a top hat vertical mass distribution? How was the MER and wind components perturbed?

L335 What was the start time of the GFS forecast that was used?

L375 Change "on this metrics" to "in this metric"

L376 Change "metrics" to "metric"

Figure labels seem to switch between capitals and lower case – these should be consistent.

---

## Author Comment (AC1)

**Response to the reviewers — Article ACP-2021-747**

Title: Data Assimilation of Volcanic Aerosols using FALL3D+PDAF

We thank the reviewers for their constructive comments, which have allowed us to improve the quality of the manuscript. We have addressed the comments and incorporated your suggestions in the revised manuscript. Our focus in the revised manuscript was to produce a self-contained work. The reader no longer needs to read the Appendix to understand the methodology. Appendix was shortened and part of it moved to a new section (Section 2) including the essential definitions and concepts. In the following we provide a detailed response addressing your comments point by point. Our responses are written following each comment. All page and reference numbers in our response are based on the revised manuscript. The line and reference numbers mentioned in the reviewers' comments are kept intact and are based on the original manuscript. Text modified or added to the manuscript is given in this format: added text. Removed text is given in this format: removed text. We hope that you find the following response satisfactory.

Sincerely,

L. Mingari, A. Folch, A.T. Prata, F. Pardini, G. Macedonio, and A. Costa

**Reviewer 1**

**General comments**

**Reviewer Point P 1.1** — More clarity is needed about how the details of the algorithm. In my view Figure 1 is nice but doesn't really help the reader understand what is actually being done. For example, even after the reading the whole paper it was not 100% clear to me how the ensembles were generated at each cycle. Do you initialize an ensemble of dispersion models using prior uncertainty estimates at t0 and compute analysis at t1, then use analysis at t1 to re-initialize the dispersion model and propagate to t2 and so on? That would mean that the initialization at the first step (volcanic source?) is quite different from initialization at subsequent steps (distal?). Please provide more concrete details so the reader doesn't need to guess.

**Reply**: We provide more concrete details on how the ensemble is generated and initialised in Section 2 and Section 3. In addition, a new figure (Fig. 1) was included to illustrate this point (see also Fig. 1 in this document). For each assimilation cycle, dispersion models are initialised using analyses at time t=t1 (actually they are restarted). These initial states are evolved in time up to time t=t2, where observations are available. The forecast at time t2 is ingested by the assimilation module to generate a new analysis at t=t2. Dispersion models are restarted at time t=t2 again using the analyses and the cycle is repeated. Note that the model is restarted from the observation time in each cycle and, therefore, initialisations are quite different from the first time t=t0. Actually, models are initialised from a zero initial concentration fields at the first time (t0). For example, we added the following sentence in Section 3:

Initially, model parameters, such as emission source parameters (ESP), and input data (e.g., meteorological fields) are sampled from a given Probability Density Function (PDF) in order to define an ensemble of model instances. In the first step, initial model conditions are defined through a set of state vectors:  $\{\vec{x}_i : i = 1, 2, ..., m\}$ , being m the ensemble size. Initial conditions can be arbitrarily defined (e.g., using data insertion). However, in this paper simulations are assumed to be started from a zero initial concentration  $(\vec{x}_i = 0)$ .

For each assimilation cycle, the analysis step requires a background ensemble  $\{\vec{x}_i^b: i = 1, 2, \ldots, m\}$ . The background states are produced by means of a forward model by evolving the ensemble of system states until a time with valid observations. At this point, a dataset of observations (including error observations) are incorporated to produce an ensemble of analyses  $\{\vec{x}_i^a: i = 1, 2, \ldots, m\}$ . The corresponding analysis for each ensemble member is used as the model initial condition for the next cycle and the forward model is restarted from the observation time. Finally, the assimilation cycle is repeated until the end of the simulation. It should be noted that model parameters are defined before simulation starts and these parameters are not resampled during subsequent assimilation cycles.

**Reviewer Point P 1.2** — On reading, it feels like the appendix was originally part of an earlier chapter. I would suggest that the authors either perhaps shorten the appendix and then insert it at an earlier stage as part of the methodology section or make an effort to make sure that discussion in Section 2 is self-contained and does not require the reader to read the appendix first. Some of the specific comments below are related to this issue.

**Reply**: As suggested by Reviewer 1, Appendix was shortened and part of it moved to a new section (Section 2) including essential concepts and definitions required to have a self-contained manuscript. The reader no longer needs to read the Appendix to understand the methodology.

**Specific comments**

**Reviewer Point P 1.3** — Introduction: There is a substantial body work on 'inverse modelling' methods using satellite retrievals of volcanic ash that has not been mentioned. See for example list of citations in Zidikheri, Meelis J., and Chris Lucas. "Improving Ensemble Volcanic Ash Forecasts by Direct Insertion of Satellite Data and Ensemble Filtering." Atmosphere 12.9 (2021): 1215. It would also be useful to mention what the DA method in this manuscript can do that these other approaches cannot do given that these methods also use observations to improve the forecasts.

**Reply**: As suggested by Reviewer 1, previous works dealing with inversion techniques are now mentioned in the Introduction. The following paragraph was included: Numerous attempts have been made to determine the eruptive source using inverse modelling techniques and satellite retrievals (e.g. Eckhardt et al., 2008; Kristiansen et al., 2010; Zidikheri and Lucas, 2020, 2021a). Typically, inversion techniques consider a simple formulation of the source term suitable to represent a single discrete eruptive event. However, multi-phase volcanic eruptions with complex emission patterns and varying temporal and spatial scales cannot be described in terms of just a few source parameters.

Figure 1: Diagram of the modelling workflow used by FALL3D+PDAF when data assimilation (DA) is enabled. Assimilation is performed by means of an ensemble-based DA technique based on a sequential scheme.

In cases where eruption source parameters are highly uncertain, data insertion becomes an interesting alternative to include information from satellite retrievals in numerical models (Wilkins et al., 2015, 2016b,a; Prata et al., 2021). In this case, instead of defining the volcanic source, numerical models are initialised directly from an initial state derived from satellite observations. Unfortunately, satellite retrievals also contain errors and missing data because of the limitations related to retrieval methods and measurement techniques. The inclusion of retrievals errors in numerical models is one of the major drawbacks of data insertion since errors will be propagated forward in time.

**Reviewer Point P 1.4** — Lines 100-105: "Background error covariance" is mentioned in Line 105 but it wouldn't be clear to readers unfamiliar with DA methods what the word "background" is referring to. It would be helpful to define "background forecast" (= "a priori forecast") earlier in the paragraph. Might also be useful to mention why the error covariance is important in DA.

**Reply**: A new section (Section 2 in the revised manuscript) was included to address these issues.

**Reviewer Point P 1.5** — Line 128: Last sentence of paragraph is hard to understand. What do you mean "filter operations are performed exclusively by ranks...". What "ranks"?

**Reply**: Ensemble modelling requires multiple instances of FALL3D running in parallel. In turn, each FALL3D instance or model task launches multiple MPI processes. However, the analysis step don't use all MPI processes. Only MPI processes corresponding to a single model task (referred to as the master model task) are used to produce the analysis ensemble. This is illustrated by Fig. 2. We replaced rank by MPI process to clarify this point. In addition, the corresponding paragraph has been rephrased for clarity:

The FALL3D+PDAF system can be run in parallel and supports online-coupled DA, enabling the workflow to be executed in a single step and with an efficient data transfer management through parallel communications. This avoids the creation of extremely large files that would be required to store the full system state in case of an off-line approach. The implementation uses a two-level parallelisation scheme based on MPI (Message Passing Interface) and can benefit from high-performance computing (HPC) resources. The two-level parallelisation scheme is sketched in Fig. 2. During the ensemble forecast phase, m instances of FALL3D, referred to as model tasks, run concurrently as an embarrassingly (or perfectly) parallel workflow to evolve the member states in time (level 1). In other words, the problem is separated into a number of parallel tasks running independently that require no communication or dependency between ensemble members. In turn, each model task is executed by a single parallel instance of FALL3D, which uses a threedimensional domain decomposition with  $n_x$ ,  $n_y$ , and  $n_z$  sub-domains along each direction (level 2). Consequently, the ensemble forecast requires a total of  $m imes n_x imes n_y imes n_z$  MPI processes. Multiple intra-member (level 1) communications are required during each assimilation step in order to collect and distribute the state vectors between different parallel tasks. Specifically, model tasks communicate with the master model task (i.e., the 1st model task in Fig. 2) during the analysis stage and filter operations required to produce the analyses are performed exclusively by the MPI processes corresponding to the master model task.

**Reviewer Point P 1.6** — Line 134: Sentence stating that LETKF is "more realistic for volcanic ash" than ETKF might need a reference (or explain why you think this would be the case). Also, this statement is rather puzzling given that you state in the abstract that LETKF didn't work very well. I think a summary of the differences between ETKF and LETKF might be needed here – including a brief discussion of the need for localisation in ensemble DA methods in general. Many readers will probably not have the time or inclination to read the appendix in detail even those details are available there. See also General Comment #2.

**Reply**: We added this phrase (last paragraph in Section 2) to justify our statement: LETKF is a more general and powerful approach as ETKF represents a particular case of LETKF in which the localisation radius is large, i.e.,  $L_R \rightarrow \infty$ . This work focuses exclusively on the LETKF technique, which provides more realistic results than its global counterpart ETKF for volcanic aerosols, as shown in Sect. 4.1.1. In Section 4.1.1 we further support this idea by means of Fig. 6a, where it

is possible to verify that the assimilation performance degrades for large  $L_R$ .

**Reviewer Point P 1.7** — Line 146: "range" – is this is the localisation radius? "inflation factor" – needs explanation.

**Reply**: A new section (Section 2 in the revised manuscript) was included to introduce these concepts. In order to define *range* we added: The localisation radius is denoted by  $L_R$  and referred to as local radius or local range throughout this work.

**Reviewer Point P 1.8** — Lines 164-166: I didn't really understand this explanation for why the ensemble forecast prior PDF forecast would be skewed. Isn't the skewness just a consequence of the way the prior ensemble is constructed? Could not in principle the prior uncertainty be sampled in such a way so as to yield a more symmetric distribution?

**Reply**: The state variable for atmospheric dispersion models, e.g., mass concentration (C), is a non-negative variable. The constraint imposed by  $C \ge 0$  is an intrinsic property of this problem that necessarily leads to skewed distributions. This is especially evident for distributions close to zero since the impossibility of having negative concentrations generates asymmetric distributions. Consequently, regardless of the sampling strategy used, we expect high skewness values as long as distributions satisfy  $\mu/\sigma \ll 1$  (i.e., when the distribution is close to zero), as found in Fig. 3a. In principle, however, a different sampling strategy could yield more symmetric distributions when  $\mu/\sigma \gtrsim 1$ . However, low mean-to-sigma ratios are extremely frequent in VATD models because, in general, a large fraction of the computational domain is not affected (or is affected with a low probability) by volcanic aerosols (this may not be valid for other atmospheric aerosols).

**Reviewer Point P 1.9** — Line 203: the overbar needs explanation (ensemble mean?)

**Reply**: You are right, it refers to the ensemble mean. The Eq. (1) was added to introduce this notation.

**Reviewer Point P 1.10** — Lines 212-216: Is there a reason for focussing on SO2 rather than volcanic ash retrievals here? Ash concentrations (rather than SO2) are of more interest in practical applications.

**Reply**: In this work, we focused on the assessment of the data assimilation method. The use of SO2 retrievals allowed us to make a direct comparison with previous simulations of the SO2 plume from Raikoke's eruption based on FALL3D (Prata et al., 2021). In fact, as Prata et al. (2021) employed a data insertion approach using the same SO2 retrieval method, we considered it a good benchmark study to show the benefits of our methodology. However, there is no essential reason for focussing on SO2 rather than ash retrievals.

Satellite retrievals of volcanic ash for this eruption have also been published recently (e.g., Muser

et al., 2020). This dataset seems to be appropriate for data assimilation purposes. We are currently working on generating our satellite retrievals for volcanic ash and we plan to assimilate volcanic ash in a future work as well. References:

Prata et al., 2021; https://doi.org/10.5194/gmd-14-409-2021 Muser et al., 2020; https://doi.org/10.5194/acp-20-15015-2020

**Reviewer 2**

**Specific comments**

**Reviewer Point P 2.1** — The introduction could benefit from the addition of a discussion of inversion modelling that can be used to constrain emission rates and plume height. Also, there is no mention of data insertion which is the simplest form of data assimilation.

**Reply**: As suggested by Reviewer 2, previous works dealing with inversion techniques and data insertion are now mentioned in the Introduction. The following paragraph was included: Numerous attempts have been made to determine the eruptive source using inverse modelling techniques and satellite retrievals (e.g. Eckhardt et al., 2008; Kristiansen et al., 2010; Zidikheri and Lucas, 2020, 2021a). Typically, inversion techniques consider a simple formulation of the source term suitable to represent a single discrete eruptive event. However, multi-phase volcanic eruptions with complex emission patterns and varying temporal and spatial scales cannot be described in terms of just a few source parameters. In cases where eruption source parameters are highly uncertain, data insertion becomes an interesting alternative to include information from satellite retrievals in numerical models (Wilkins et al., 2015, 2016b,a; Prata et al., 2021). In this case, instead of defining the volcanic source, numerical models are initialised directly from an initial state derived from satellite observations. Unfortunately, satellite retrievals also contain errors and missing data because of the limitations related to retrieval methods and measurement techniques. The inclusion of retrievals errors in numerical models is one of the major drawbacks of data insertion since errors will be propagated forward in time.

**Reviewer Point P 2.2** — The authors motivate the study by citing the impacts of volcanic ash on aviation, but the second set of experiments focus on the assimilation of sulphur dioxide. What was the reason for this? There are satellite retrievals of ash available for this eruption or are they too patchy? Is the fact that the satellite can only "see" the distal ash plume a problem?

**Reply**: In this work, we focused on the assessment of the data assimilation method. The use of SO2 retrievals allowed us to make a direct comparison with previous simulations of the SO2 plume from Raikoke's eruption based on FALL3D (Prata et al., 2021). In fact, as Prata et al. (2021) employed a data insertion approach using the same SO2 retrieval method, we considered it a good benchmark study to show the benefits of our methodology. However, there is no essential reason for focussing on SO2 rather than ash retrievals.

Satellite retrievals of volcanic ash for this eruption have also been published recently (e.g., Muser et al. 2020). This dataset seems to be appropriate for data assimilation purposes. We are currently working on generating our satellite retrievals for volcanic ash and we plan to assimilate volcanic ash in a future work as well.

Regarding your last question, the DA method works even if only distal observations are available. In fact, in this work we showed that it is possible to reconstruct the *true state* by assimilating an incomplete dataset of observations. However, the quality of the analysis will potentially be degraded in proximal areas if no observation are available there. References:

Prata et al., 2021; https://doi.org/10.5194/gmd-14-409-2021 Muser et al., 2020; https://doi.org/10.5194/acp-20-15015-2020

**Reviewer Point P 2.3** — I am unsure how Figure 1 enhances the readers understanding of the method – how is the ensemble constructed? Also, does the assimilation of the satellite retrievals take into account their uncertainty?

**Reply**: We provide more concrete details on how the ensemble is generated and initialised in Section 2 and Section 3. In addition, a new figure (Fig. 1) was included to illustrate this point (see also Fig. 1 in this document). For example, we added the following sentence in Section 3: Initially, model parameters, such as emission source parameters (ESP), and input data (e.g., meteorological fields) are sampled from a given Probability Density Function (PDF) in order to define an ensemble of model instances. In the first step, initial model conditions are defined through a set of state vectors:  $\{\vec{x}_i : i = 1, 2, ..., m\}$ , being m the ensemble size. Initial conditions can be arbitrarily defined (e.g., using data insertion). However, in this paper simulations are assumed to be started from a zero initial concentration  $(\vec{x}_i = 0)$ . Observation errors are required in order to compute the observation error covariance matrix (assumed diagonal):  $\mathbf{R} \in \mathbb{R}^{p \times p}$ , where p is the number of observations to be assimilated. This

is clarified now in Section 2.

**Reviewer Point P 2.4** — L150 You refer to something in the appendix – does this part of the appendix to be worked into the main body of the text?

**Reply**: We agree with Reviewer 2. Appendix was shortened and part of it moved to a new section (Section 2) including essential concepts and definitions required to have a self-contained manuscript. The reader no longer needs to read the Appendix to understand the methodology.

**Reviewer Point P 2.5** — The authors show that the prior pdf associated with the ensemble forecast tends to be skewed possibly leading to the unrealistic posterior estimate as the Gaussian assumption in Kalman filter theory is not satisfied. Can the prior pdf be modified by different parameter sampling strategy or constructing the prior in a different way?

**Reply**: Gaussian anamorphosis methods (Bertino et al., 2003) aim to construct transformations to turn the state vector into a Gaussian vector. For example, a nonlinear function can be applied to the cumulative pdf in order to make it Gaussian (Chilès and Delfiner, 2012). However, the drawback of this method is that the transformation can introduce significant nonlinearities if distributions are highly skewed.

References:

Bertino et al., 2003; https://doi.org/10.1111/j.1751-5823.2003.tb00194.x Chilès, J. P., & Delfiner, P. (2012). Geostatistics: Modeling spatial uncertainty.

**Reviewer Point P 2.6** — Figure 6 and the comparison of observations, free run and analysis of the Raikoke eruption – are the distributions shown for the free run and analysis the ensemble means? In panel G, is the southern branch of ash missing due to the presence of meteorological cloud?

**Reply**: You are right, they are ensemble means. This is clarified in the caption of Fig. 7 now. The southern branch of SO2 missing in panel G is probably related to a limitation of the retrieval method. The retrieval is based on the strong absorption of SO2 near the 7.3  $\mu m$  wavelength and is generally only sensitive to upper-level SO2 due to the masking effect of water vapour absorption at lower levels in the atmosphere (see Section 3.6). Note that, according to our simulations, the southern branch is a low-level cloud (Fig. 10c).

Finally, the southern branch could be detected by Muser et al. (2020) according to the SO2 mass loading measurements derived from TROPOMI observations. Reference:

Muser et al., 2020; https://doi.org/10.5194/acp-20-15015-2020

**Reviewer Point P 2.7** — Figure 9 I really like panel C as you can see the ascent in the cyclone. It would be nice to see a similar plot for the free running ensemble. This might help explain the large differences between the free running simulation and observations seen in Figure 10. Can FALL-3D represent diabatic heating which can also cause ascent?

**Reply**: Unfortunately, it is not possible for us to represent the diabatic heating since we are using an offline modelling approach here. However, previous studies have shown that an online treatment of the plume dynamics has an important impact on simulations due to the effect of aerosol-radiation interaction (Bruckert et al, 2021). Reference:

Bruckert et al, 2021; https://doi.org/10.5194/acp-2021-459

**Reviewer Point P 2.8** — Assimilation is expensive - can anything be gained/lost from more/less frequent assimilation?

**Reply**: Ideally, assimilation should be as frequent as possible. However, ensemble spread decreases at each assimilation time, which is compensated by the ensemble variability introduced

during each forecast period. If assimilation frequency is too high, the ensemble could "collapse". The ensemble spread should be greater than the analysis error in order to avoid under-dispersive ensembles. We shown that with an assimilation frequency of 3h the ensemble spread remained above the RMSE during each assimilation cycle (see Fig. 6b and discussion in Sect. 3.1.1). At least in this case, the optimal assimilation frequency was 3h.

In principle, more advanced strategies could be employed to overcome this limitation in the frequency of assimilation. For example, additional observations could be included during each assimilation step without the need to reduce the frequency of assimilation by assimilating measurements within a given period of time or observation window. This could have some impact on the quality of forecasts and this is an interesting topic for future research.

**Technical comments**

**Reviewer Point P 2.9** — L2 and 25 Unsure what is meant here by "infrastructures"

**Reply**: Fixed. Replaced infrastructures by buildings

**Reviewer Point P 2.10** — L67 Change "enabled to quantify model uncertainties" to "enabled the quantification of model uncertainties"

**Reply**: Fixed

**Reviewer Point P 2.11** — L123 Unsure what is meant here by "embarrassingly (or perfectly)". The same on L244.

**Reply**: This is clarified now in Section 3.1. We added: In other words, the problem is separated into a number of parallel tasks running independently that require no communication or dependency between ensemble members.

**Reviewer Point P 2.12** — L136 Is there a reference for the "realistic results" you mention?

**Reply**: We added this phrase (last paragraph in Section 2) to justify our statement: LETKF is a more general and powerful approach as ETKF represents a particular case of LETKF in which the localisation radius is large, i.e.,  $L_R \rightarrow \infty$ . This work focuses exclusively on the LETKF technique, which provides more realistic results than its global counterpart ETKF for volcanic aerosols, as shown in Sect. 4.1.1. In Section 4.1.1 we further support this idea by means of Fig. 6a, where it is possible to verify that the assimilation performance degrades for large  $L_R$ .

**Reviewer Point P 2.13** — L146 Is the local range referred to here the same as  $L_R$ ? Can you expand on the inflation factor that is referred to?

**Reply**: You are right,  $L_R$  is the local range. We added a new section (Section 2) to expand on these concepts. For example, we added: The localisation radius is denoted by  $L_R$  and referred to as local radius or local range throughout this work.

Reviewer Point P 2.14 — Table 2 caption last line – change if to of

Reply: Fixed

Reviewer Point P 2.15 — Table 2 WRF-ARW needs to be defined

Reply: Fixed

**Reviewer Point P 2.16** — Equation 4 What does tr mean? Is n the number of ensemble members?

**Reply**: This is the trace of a square matrix. This is clarified now: and  $tr(P_e)$  denotes the trace of  $P_e$

**Reviewer Point P 2.17** — Table 3 Change grid size to resolution, Domain size to number of grid points, expand TGSD

**Reply**: Fixed

**Reviewer Point P 2.18** — Figure 3 This seems to be a very complex column height profile. Is it representative of what might be used in operations?

**Reply**: These profiles are intended to be representative of a real eruptive scenario and are used to construct the *true state*. However, the true state is unknown in operations (only satellite observations are available). We only need the true state to quantitatively evaluate the performance of the data assimilation method.

**Reviewer Point P 2.19** — L279 It would be nice to remind the reader that it is a 36-hour forecast being performed here

Reply: Done

**Reviewer Point P 2.20** — L280 Why is the flow rate fixed? Could this also be perturbed or determined from the perturbed plume height?

**Reply**: The mass flow rate, MFR, can be perturbed or computed from column height, H (regardless of whether H was perturbed or not). However, we decided to assume a fixed MFR in order to avoid correlations between the background ensemble and the true state (MFR was computed from H in the definition of the true state).

**Reviewer Point P 2.21** — L296 Is there a reference for the "notorious degradation"?

**Reply**: This is shown in Fig. 6a. We added the reference: (see Fig. 6a).

**Reviewer Point P 2.22** — L326 What was the motivation for using a top hat vertical mass distribution? How was the MER and wind components perturbed?

**Reply**: We used a top-hat distribution because lidar measurements show a very localised distribution of aerosols over upper layer of atmosphere (e.g., Muser et al., 2020). MER and horizontal wind components were perturbed assuming uniform PDF's and using a Latin Hypercube Sampling strategy. MER was perturbed from a central value of  $2 \times 10^5 \ kg/s$  with a perturbation range of  $\pm 20\%$ . The U- and V-components of wind were independently perturbed assuming a perturbation range of  $\pm 25\%$ . In this case, central values were obtained from the GFS forecast (see Table 2 for further details).

Reviewer Point P 2.23 — L335 What was the start time of the GFS forecast that was used?

Reply: We used GFS initialised on 21 June 2019 at 18:00 UTC.

Reviewer Point P 2.24 — L375 Change "on this metrics" to "in this metric"

Reply: Fixed

Reviewer Point P 2.25 — L376 Change "metrics" to "metric"

Reply: Fixed

**Reviewer Point P 2.26** — Figure labels seem to switch between capitals and lower case – these should be consistent.

Reply: Fixed

---

## Author Response (AR2)

Dear Farahnaz Khosrawi,

Thank you for accepting our manuscript for final publication in ACP.

I appreciate your valuable comments. Please, find my answers below:

**Comment 1:**

P1, L20: But what is actually the best approach? Later in the manuscript you write that also EnKF does not work. Is the approach you present here the only option to treat volcanic aerosol in data assimilation or are there also other (better) options?

Reply: The LETKF method used in this work belongs to the EnKF (Ensemble Kalman Filter) family of methods. Despite the limitations of this specific method, LETKF has shown to be a promising alternative for assimilation of volcanic data. Our findings show that a significant improvement of the forecast quality can be achieved with this method. Therefore, we consider that this method actually works.

However, LETKF results in suboptimal filter performance and there is room for further improvement. In Sect. 5, we discuss potential alternative methods to be investigated in future works. Here, we suggest focussing on non-traditional EnKF variants. In particular, we mention the ensemble Kalman filtering strategy proposed by Bishop (2016) for highly skewed non-negative uncertainty distributions.

However, further work needs to be done to establish whether these alternative methods are better.

**Comment 2:**

P3, L56: retrievals → retrievals

Reply: We cannot see any error in this sentence. Please, could you clarify "retrievals->retrievals"?

**Comments 3:**

P12, Eq. 7: Please check the formula. I think a "2" in the exponent is missing to end up with the correct calculation.

Reply: Thank you for noticing this error. We have fixed it now.

**Comments 4:**

P13, Eq 8: Same here as for Eq. 7.

Reply: Fixed.

**Comments 5:**

P16, L362: "Fig." missing before "5a", "5b", "5c" and "5d".

Reply: Fixed.

**Comments 6:**

P20, L427: space between "LETKF" and "(sqrt)" missing.

Reply: Fixed.

**Comments 7:**

P20, L438: unit should be in upright fond.

Reply: Fixed.

**Comments 8:**

P25, L493: ......by a complex plume dynamics" → mixture of singular and plural. Chose either singular or plural.

Reply: Fixed. We removed "a". Replaced: "by a complex plume dynamics" by "by complex plume dynamics"